# ANALYSIS OF HIGH-ORDER INTERACTIONS IN SHAPLEY VALUE FOR MODEL INTERPRETATION

## ABSTRACT

The Shapley value is a fundamental game-theoretic framework for allocating a utility function's output among participating players, and is commonly interpreted as the expected marginal contribution under random coalitions. However, when applied to complex functions such as deep neural networks, this expected marginal contribution implicitly aggregates higher-order interaction effects, which can obscure the true contribution of features. In this study, we derive a generalized decomposition of the Shapley value that expresses it as a sum of interaction terms of arbitrary order, making explicit how higher-order interactions are incorporated within marginal contributions. We also provide an unbiased estimator for our representation via permutation sampling, enabling practical computation. We further show that when interaction effects vary substantially across contexts, these embedded higher-order terms can lead to misleading attributions for model interpretation. Our theoretical analysis and empirical evaluations demonstrate that variance in lower-order interactions reliably signals the presence of hidden higher-order structure, providing a principled criterion for when such interactions should be explored. This interaction-based perspective clarifies when the Shapley value becomes unreliable and offers new guidance for interpreting model behavior.

## 1 INTRODUCTION

The Shapley value is a fundamental solution concept in cooperative game theory that fairly allocates the total utility of a game among players (Shapley, 1953). Owing to its strong theoretical foundation, it has also become the basis of many feature attribution methods in machine learning, where the model output is treated as the utility of a cooperative game and contributions are distributed among input features. This game-theoretic perspective has established the Shapley value as one of the most influential tools in explainable AI (XAI) (Lundberg & Lee, 2017a; Sundararajan et al., 2017; Ghorbani & Zou, 2020; Lundberg et al., 2020; Wang et al., 2021; Rozemberczki et al., 2022).

The standard Shapley value is interpreted as the expected *marginal contribution* of each player under random coalitions. Yet each marginal contribution is inherently shaped by *interaction effects* between the target player and other coalitions (VanderWeele, 2015; Egami & Imai, 2019; Grabisch & Roubens, 1999; Chang et al., 2025). Consequently, simple expectation often fails in complex functions such as Deep Neural Networks (DNNs), where high-order interactions are pervasive and ignoring these interactions can cause the algorithm to overlook critical cooperative structures and even yield misleading interpretations.

In this work, we formalize this perspective by proving that the Shapley value can be expressed as a decomposition of the characteristic function into interaction terms of arbitrary order, where each term is evenly distributed among the players involved. This representation reveals the internal structure of Shapley's expectation-based formulation: lower-order effects implicitly subsume all higher-order interactions, explaining why context-sensitive effects may be obscured in standard attributions. Our theoretical results generalize the classical dividend decomposition (Harsanyi, 1982; Dehez, 2017) and make explicit how higher-order dividends are embedded within the Shapley value. We further show that permutation-based sampling yields an unbiased estimator of the $k$-th order representation of the Shapley value, enabling practical computation (Castro et al., 2009).

Within this $k$-th order representation, our analysis clarifies when the Shapley value becomes unreliable: when interaction terms fluctuate substantially across coalitions, their expectation can mask in-

dispensable higher-order structure and yield near-zero attributions for meaningful features. Through theoretical case studies and empirical evaluations on DNNs, we show that large variance in low-order interactions reliably signals such hidden higher-order structure. This motivates a variance-based strategy for prioritizing coalitions in higher-order exploration, which is particularly valuable in high-dimensional deep learning settings where interaction patterns are sparse and exhaustive evaluation is infeasible.

In summary, our work revisits the Shapley value by making its embedded interaction structure explicit and by identifying when its expectation-based formulation fails to capture true feature importance. Our decomposition reveals how higher-order interactions are implicitly aggregated within marginal contributions, providing a principled explanation for these failures. To extend these insights, we introduce the High-Variance Effect (HIVE) framework, which uses variance as a criterion for guiding higher-order exploration while pruning uninformative coalitions. This variance-guided strategy yields a scalable approach for uncovering meaningful higher-order interactions and offers new directions for interpreting how modern deep neural networks organize interaction structure.

## 2 SHAPLEY VALUE

**Notation.** For convenience, we follow the simplified notations in Grabisch & Roubens (1999); Fujimoto et al. (2006). For singletons, we omit braces and write $v(i), T \cup i, T \setminus i$ instead of $v(\{i\}), T \cup \{i\}, T \setminus \{i\}$. Similarly, for multiple elements, we use $ij, ijk$ instead of $\{i, j\}, \{i, j, k\}$ when it is clear. The cardinalities of subsets $S, T, R \cdots$ are typically denoted by the corresponding lowercase letters $s, t, r, \cdots$.

**Shapley value.** In cooperative game theory, a *cooperative game* consists of a set of players $N = \{1, \ldots, n\}$ and a characteristic function $v : 2^N \to \mathbb{R}$ (also called utility function) that maps each coalition $S \subseteq N$ to the utility $v(S)$. The player $i$'s *marginal contribution* (also called *effect*) measures the added value when player $i$ joins an existing coalition $S$, $\Delta_i v(S) := v(S \cup i) - v(S)$. It can be extended to the group marginal contribution, $v(S \cup R) - v(S)$.

The *Shapley value* is one of the solution concepts to fairly allocate the utility to individual players with specific axioms in a cooperative game. The solution assigns to each player a payoff equal to the expectation of $\Delta v_i(S)$ over all coalitions $S \subseteq N \setminus i$ (Shapley, 1953; Monderer & Samet, 2002):

$$\phi_i(v) = \sum_{S \subseteq N \setminus i} \frac{1}{n} \binom{n-1}{s}^{-1} \left[ v(S \cup \{i\}) - v(S) \right]. \tag{1}$$

The Shapley value can also be represented as the expectation over all permutations of players, which provides a more efficient approximation in practice (Castro et al., 2009). Let $\Pi(N)$ be the set of all permutations of $N$. For $\pi \in \Pi(N)$, the set $\pi^i$ denotes the set of players that precede $i$ in $\pi$. Then, the Shapley value is the same as follows:

$$\phi_i(v) = \frac{1}{n!} \sum_{\pi \in \Pi(N)} \left[ v(\pi^i \cup i) - v(\pi^i) \right]. \tag{2}$$

**Harsanyi dividend.** In a different perspective, instead of marginal contributions, the Shapley value can be decomposed into *dividends* of all possible coalitions (Harsanyi, 1982; Dehez, 2017). The dividend $\alpha_R(N, v)$ is defined as follows:

$$\alpha_R(N, v) = \sum_{T \subseteq R} (-1)^{r-t} v(T). \tag{3}$$

It measures the pure effect of coalition $R$ that cannot be explained by its subcoalitions. $\alpha_R(N, v)$ is often simplified as $\alpha_R$ when the context of $N, v$ is clear. This definition provides unique representations of the characteristic function and the Shapley value in the following forms:

$$v(S) = \sum_{R \subseteq S} \alpha_R, \quad \phi_i(v) = \sum_{R \subseteq N, i \in R} \frac{1}{r} \alpha_R. \tag{4}$$

## 3 INTERACTION IN SHAPLEY VALUE

The classical formulation of the Shapley value attributes payoffs to players by averaging their individual effects across coalitions. However, in modern applications, particularly when applied to complex functions like DNNs, it becomes crucial to understand the interactions among players (i.e., model features) beyond the individual level. Recent studies have found that such interactions sparsely capture meaningful semantic concepts (Deng et al., 2021; Li & Zhang, 2023; Ren et al., 2023; Zhou et al., 2024; Kang et al., 2025), which suggests that analyzing interactions provides a more faithful explanation of complex models than focusing solely on individual features. In this section, we (a) demonstrate how higher-order *interaction effects* are implicitly embedded in lower-order interactions, (b) show efficient estimation of interactions via permutation sampling, and (c) provide an interpretation of the Shapley value with explicit reformulation with respect to arbitrary-order interaction terms.

### 3.1 INTERACTION EFFECTS

In the two-player case, the interaction effect between $i$ and $j$ with a given player set $T \subseteq N \setminus ij$ indicates the discrepancy in the effect of one variable when the other is present.

$$\begin{aligned}
\Delta_{ij}v(T) = \Delta_j\big[\Delta_i v(T)\big] &= \Delta_i v(T \cup j) - \Delta_i v(T) \\
&= v(T \cup ij) - v(T \cup i) - v(T \cup j) + v(T).
\end{aligned} \tag{5}$$

A positive interaction indicates synergistic effects from cooperation, while a negative value implies redundancy or conflicts between players (Fujimoto et al., 2006; Fumagalli et al., 2024; Chang et al., 2025). The definition can be extended to any subset by recursively computing the discrepancy.

**Definition 1** (interaction). The interaction of coalition $R \subseteq N$ for a given coalition $T$ is:

$$\begin{aligned}
\Delta_R v(T) &= \Delta_i[\Delta_{R \setminus i} v(T)], \quad \forall i \in R \\
&= \sum_{S \subseteq R} (-1)^{r-s} v(S \cup T).
\end{aligned} \tag{6}$$

In particular, we call the $k$-th order interaction for the case $|R| = k$. This term follows the causal interaction in causality literature that evaluates the interaction effects among variables by intervention on target variables (VanderWeele, 2015; Egami & Imai, 2019; Janzing et al., 2020), i.e., the additional effect of the coalition beyond the sum of all lower-order interactions. With convention $\Delta_\emptyset v(T) := v(T)$, it satisfies the following equation:

$$\Delta_R v(T) = v(R \cup T) - \sum_{S \subset R} \Delta_S v(T). \tag{7}$$

Note that the term 'interaction' in this study indicates *causal interaction* to understand implicit interaction effects behind the Shapley value, not *interaction index* in game theory literature, which provides a generalized allocation framework for a subset of players. This interaction equation also follows the structure of *discrete derivative*, which computes the function change by inclusion-exclusion (Fujimoto et al., 2006). The Harsanyi dividend in Equation (3) is the special case of interaction when $T = \emptyset$ (Dehez, 2017). That is, $\alpha_R = \Delta_R v(\emptyset)$.

### 3.2 INTERACTION DECOMPOSITION

In this section, we introduce a new formulation of the Shapley value using $k$-th order interaction terms. We first explain how higher order interactions are implicitly embedded in the marginal contribution. Consider the marginal contribution $\Delta_i v(S)$ and a random permutation $\pi \in \Pi(S)$. Let $[\pi]_t$ be the subset of players up to the $t$-th player in the ordering $\pi$, where $[\pi]_0 := \emptyset$ and $[\pi]_s := S$. $\pi^R$ denotes the set of players in $\pi$ that precede all players in $R$. By definition, $\Delta_i v(S) - \Delta_i v(\emptyset)$ can be decomposed into a consecutive summation of 2nd-order interactions according to the permutation. Each 2nd-order interaction can be decomposed into a summation of 3rd order interactions. By recursively applying this decomposition for all permutations, we obtain the following lemma.

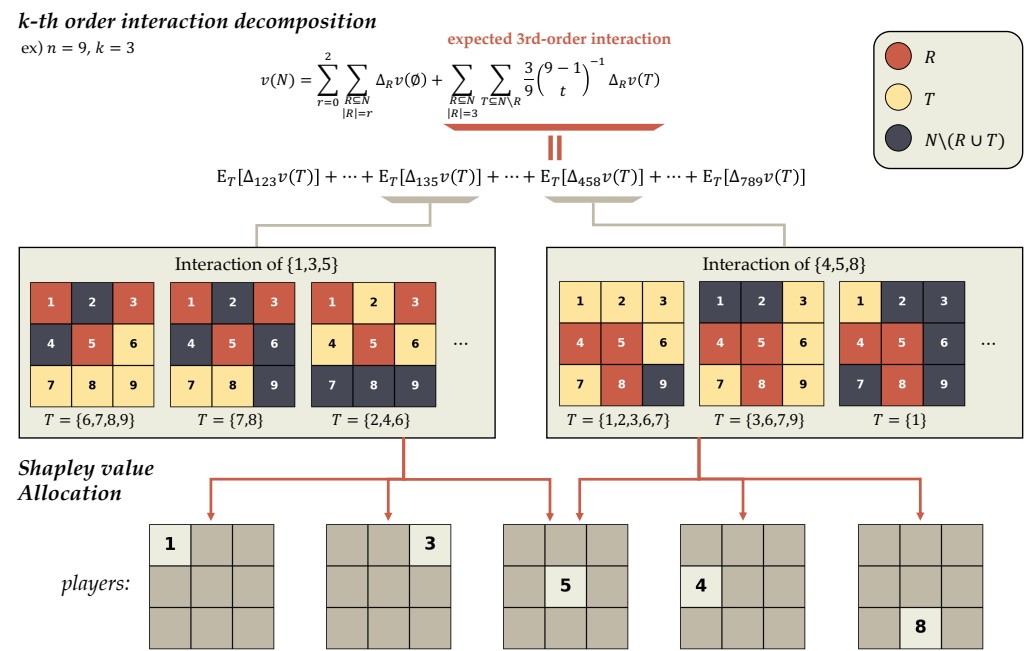

Figure 1: An illustration of $k$-th order Interaction Decomposition in Shapley value for $k = 3$. For a given coalition $R$ and context $T$, each interaction term involving $R$ is divided evenly among the constituents of $R$.

**Lemma 1** ($k$-th order interaction in $\Delta_i v(S)$). *For a permutation $\pi \in \Pi(S)$, for any $t \in [0, s-k+1]$,*

$$\Delta_i v(S) = \sum_{r=0}^{k-2} \sum_{\substack{R \subseteq S \setminus [\pi]_t \\ |R|=r}} \Delta_{i \cup R} v([\pi]_t) + \sum_{\substack{R \subseteq S \setminus [\pi]_t \\ |R|=k-1}} \Delta_{i \cup R} v(\pi^R). \tag{8}$$

*Proof.* See Appendix. $\square$

Lemma 1 shows that a marginal contribution $\Delta_i v(S)$ can be expressed as a consecutive sum of interaction terms between $i$ and subsets of $S$. Since both the characteristic function $v$ and the Shapley value $\phi_i(v)$ are defined in terms of marginal contributions, they too admit representations in terms of interactions. By substituting each marginal contribution with its $k$-th order interaction expansion from Lemma 1, we obtain $k$-th order interaction decompositions of the set function $v$ and the Shapley value.

**Theorem 1** ($k$-th order interaction representation of a set function). *Given a set function $v : 2^N \to \mathbb{R}$ and a subset $S \subseteq N$, $v(S)$ can be expressed with $k$-th order interaction terms:*

$$v(S) = \sum_{r=0}^{k-1} \sum_{\substack{R \subseteq S \\ |R|=r}} \Delta_R v(\emptyset) + \sum_{\substack{R \subseteq S \\ |R|=k}} \sum_{T \subseteq S \setminus R} \frac{k}{s} \binom{s-1}{t}^{-1} \Delta_R v(T) \tag{9}$$

**Theorem 2** ($k$-th order decomposition of Shapley value). *The Shapley value can be represented in $k$-th order interactions:*

$$\phi_i(v) = \sum_{r=0}^{k-2} \frac{1}{r+1} \sum_{\substack{R \subseteq N \setminus i \\ |R|=r}} \Delta_{i \cup R} v(\emptyset) + \sum_{t=0}^{n-k} \frac{1}{n} \binom{n-1}{t}^{-1} \sum_{\substack{R \subseteq N \setminus i \\ |R|=k-1}} \sum_{\substack{T \subseteq N \setminus (i \cup R) \\ |T|=t}} \Delta_{i \cup R} v(T). \tag{10}$$

*Proof.* See Appendix. $\square$

Theorem 2 explicitly shows how higher-order interaction terms contribute to the Shapley value. An interesting observation can be made for $S = N$ in Theorem 1. Here, the weight of $\Delta_R v(T)$ equals the product of the interaction weight from Theorem 2 and the number of players involved. This equivalence shows that computing the Shapley value is equivalent to decomposing the characteristic function into interactions up to an arbitrary order and evenly distributing each interaction term among the participating players. In other words, the Shapley value can be viewed as a fair allocation of decomposed interaction effects. This interpretation is illustrated in Figure 1.

### 3.3 INTERACTION ESTIMATION VIA PERMUTATION SAMPLING

To efficiently estimate higher-order interaction terms in Theorem 2, we introduce an unbiased estimator based on permutation sampling, following the approach of Castro et al. (2009). Let $\pi_t$ denote the $t$-th player in a permutation $\pi$, and $[\pi]_t$ the subset of players up to position the $t$-th player. We define $\Delta_{i\pi_{t+1}}v([\pi]_t) = 0$ whenever $i \in [\pi]_{t+1}$ so that the term is well-defined for any player $i$ and permutation $\pi$. This leads to a simplified estimator that applies uniformly across sampled interactions. Formally, for any $k \in [2, n]$, the following unbiased estimation holds:

**Theorem 3** (estimation via permutation sampling). *The Shapley value with $k$-th order interactions can be estimated through permutation sampling:*

$$\phi_i(v) = \sum_{\substack{R \subseteq N\setminus i \\ |R| \in [0, k-2]}} \frac{1}{r+1} \Delta_{i\cup R}v(\emptyset) + \frac{1}{k-1} \sum_{t=0}^{n-k} \mathbb{E}_{\pi \in \Pi(N)}\Big[ \sum_{\substack{R \subseteq N\setminus[\pi]_{t+1} \\ |R|=k-2}} \Delta_{i\pi_{t+1}\cup R}v([\pi]_t) \Big] \quad (11)$$

*Proof.* See Appendix. $\qquad\square$

This permutation-based approach enables efficient estimation of interaction terms in practice. Unlike set-based sampling, which may produce sparse or imbalanced coverage, permutation sampling assigns equal weight to each interaction and achieves better sample efficiency. Our formulation generalizes the 2nd-order interaction estimation result introduced in Corollary 1 of Chang et al. (2025). We provide an empirical analysis of our permutation-based estimation on Appendix F.

### 3.4 INTERPRETATION AND LIMITATIONS OF INTERACTIONS IN SHAPLEY VALUE

**Summarization of higher-order interactions.** The special case when $k = n$ in Theorem 1 and 2 recovers the classical dividend-based representation of the Shapley value in Equation (4). Our results therefore generalize this interpretation by decomposing the Shapley value up to a desired order. The second term in Theorem 2 is the expected $k$-th order interaction involving the player $i$. These expectation terms implicitly encode higher-order Harsanyi dividends. For any coalition $R$, the interaction effects satisfy the classical identity (Grabisch & Roubens, 1999; Fujimoto et al., 2006)

$$\Delta_R v(T) = \sum_{S \subseteq T} \alpha_{R \cup S}. \quad (12)$$

Thus, the expected interaction in Theorem 2 becomes a weighted summarization of all higher-order dividends over supersets of $i \cup R$.

**Theorem 4** (dividends in $k$-th order interaction representation). *The Harsanyi dividend of $L \subseteq N$ is embedded in the $k$-th order interaction representation of Shapley value as follows:*

$$\phi_i(v) = \sum_{r=0}^{k-2} \frac{1}{r+1} \sum_{\substack{R \subseteq N\setminus i \\ |R|=r}} \alpha_{i\cup R} + \sum_{\substack{R \subseteq N\setminus i \\ |R|=k-1}} \sum_{\substack{L \subseteq N \\ (i\cup R) \subseteq L}} \frac{1}{k}\binom{l}{l-k}^{-1} \alpha_L. \quad (13)$$

*Proof.* See Appendix. $\qquad\square$

It reveals how Shapley's marginal contribution implicitly summarizes higher-order dividends. Averaging $\Delta_R v(T)$ at order $k$ provides a practical alternative to computing the full Harsanyi expansion,

which becomes infeasible at scale, while still reflecting the combined influence of higher-order interaction structure. The original Shapley value corresponds to evaluating this summarization at $k = 1$, thereby subsuming all higher-order effects into the expectation of first-order marginal contributions.

**Problems with expectation-based evaluation.** The limitation of this summarization is that it may suppress or distort critical interactions. When $\Delta_i v(T)$ does not heavily rely on the context $T$, the expectation is a reliable measure of the feature attribution since there is no substantial interaction effect between $i$ and features in $T$. However, when $\Delta_i v(T)$ is highly context-sensitive, which is common in complex non-additive or non-convex architectures like DNNs, the expectation collapses heterogeneous interaction effects into a single aggregate value. This can obscure the true role of the feature and lead to misleading attributions.

This issue becomes especially problematic in the presence of negative or redundant interactions (Kumar et al., 2021; Chang et al., 2025). Even in non-convex models, a relevant feature may appear irrelevant because positive contributions can be canceled out by negative interactions induced by redundancy. This cancellation can occur not only at the second order but also at higher orders: a positive pairwise interaction can flip sign when additional features participate due to higher-order negative interactions. Thus, when $\Delta_R v(T)$ varies substantially across contexts, the expectation $\mathbb{E}_T[\Delta_R v(T)]$ becomes an unreliable summary of the coalition's true influence. In such scenarios, exploring higher-order structure is essential. Rather than relying solely on expectation-based summarization, one must examine how the interaction behaves across different contexts and how different supersets activate distinctive Harsanyi dividends.

### 3.5 RELATION TO PRIOR WORK

A substantial line of work extends the Shapley value to quantify feature interactions, beginning with the Shapley Interaction Index (SII) (Grabisch & Roubens, 1999). Because SII does not satisfy efficiency, later methods such as STI (Sundararajan et al., 2020) and Faith-Shap (Tsai et al., 2023) incorporate this axiom, and n-Shapley (Bordt & von Luxburg, 2023) further unifies these formulations. All these indices reduce to the Shapley value at singleton levels and recover Harsanyi dividends at full cardinality, but differ in how they allocate higher-order effects.

Our work takes a different perspective. Rather than defining a new interaction index, we analyze how Shapley's expectation over marginal contributions is influenced by the intrinsic higher-order structure of non-additive and non-convex models. By decomposing each marginal contribution into a consecutive sum of interaction effects (Lemma 1), our formulation satisfies efficiency and reveals how Shapley-based explanations implicitly accumulate higher-order dividends. In this process, our max-order interaction term also naturally accumulates all higher-order dividends since we iteratively decompose marginal contributions from low order to high order. Thus, the second term in Theorem 2 corresponds to the aggregation of STI uniformly allocated to each feature in $i \cup R$.

Despite these connections, our work highlights that expectation-based evaluations can suppress critical interactions when discrete derivatives change sign across contexts, which is a phenomenon also noted by Shapley residuals (Kumar et al., 2021) and negative interactions in non-convex models (Chang et al., 2025). These sign cancellations can yield misleadingly small or even zero attributions, not because of the choice of interaction index, but due to the intrinsic structure of marginal contributions themselves.

Our formulation makes this issue explicit by showing exactly how higher-order discrete derivatives are embedded within the marginal contributions. Section 4 illustrates this through simple operator examples, and Section 5 demonstrates that the same phenomenon appears in deep neural networks. These observations motivate the need for principled guidance to identify higher-order coalitions exhibiting context-sensitive, non-negligible interactions. This need is particularly pronounced in modern deep models, where meaningful higher-order interactions are extremely sparse. Recent works such as SPEX and ProxySPEX (Kang et al., 2025; Butler et al., 2025) further support this view by showing that impactful interactions in large language models often arise along only a small number of coalitional pathways.

Finally, although computing high-order discrete derivatives remains expensive, our formulation lies within the Cardinal Interaction Index (CII) class, making it compatible with efficient estimators such as SHAP-IQ and SVARM-IQ (Fumagalli et al., 2023; Muschalik et al., 2024; Kolpaczki et al.,

2024). Combining our guided-exploration strategy with these estimators offers a promising path toward scalable higher-order interaction analysis.

## 4 CASE STUDY

As discussed in Section 3.4, interaction effects are implicitly embedded in the marginal contribution by expectation when computing the Shapley value. This expectation structure may lead to unexpected allocation in complex functions where indispensable high-order interactions exist. We demonstrate this concept with two example functions that are frequently used in DNNs: max functions and attention.

**Max function.** The max function selects the largest value among its inputs and is widely used in various DNNs, e.g., max pooling.

$$v(x_1, \cdots, x_5) = 4x_1 + \max(7x_2, 8x_3, 9x_4, 10x_5)$$

We set each variable $x_i$ as binary (0 or 1) to represent the participation of player $i$. We then examine the necessity of analyzing $k$-th order interactions $\Delta_R v(T)$, with a particular focus on $x_1$ and $x_5$. We can easily find that the marginal contribution of $x_1$ ($\Delta_1 v(T)$) is always 4 regardless of the coalition $T$. Thus, the Shapley value of $x_1$ (4) adequately summarizes its contribution. However, the marginal contribution of $x_5$ significantly differs depending on the coalition it joins. For instance, the marginal contribution of $x_5$ becomes 10, 3 and 1 when $T$ is $\emptyset$, $\{1\}$ and $\{1, 3, 4\}$, respectively. Due to such large variations, the expectation reflected in $x_5$'s Shapley value (3.58) does not well capture $x_5$'s coalition-specific effects, highlighting the need to account for higher-order interactions.

**Attention module.** The attention module is a widely used component in modern DNNs (Vaswani et al., 2017; Dosovitskiy et al., 2020; Ho et al., 2020). For this example, we simplify its computation structure. Specifically, we define

$$\mathbf{z} = [3x_1, 5x_2, 9x_3, 10x_4]^T$$
$$v(x_1, \cdots, x_5) = \text{softmax}(x_1, x_2, x_3, x_4)^T \mathbf{z}$$

where $\text{softmax}(x_1, x_2, x_3, x_4)_i = e^{x_i} / \sum_{j=1}^{4} e^{x_j}$ indicates the attention weight of player $i$, and $\mathbf{z}$ represents the corresponding value vectors (Vaswani et al., 2017). Similar to $x_5$ in the max function example, $x_1$ exhibits substantially different effects depending on the presence of $x_2$, $x_3$, and $x_4$. For instance, $x_1$ contributes 1.43 in isolation ($T = \emptyset$), but makes negative contributions ($-0.41$ and $-0.38$) when $T = \{3, 4\}$ and $T = \{2, 3, 4\}$. Furthermore, 2nd and higher-order interactions fluctuate across coalitions, e.g., $\Delta_{12} v(\emptyset) = -0.88$, $\Delta_{12} v(3, 4) = -0.04$, and $\Delta_{123} v(4) = 0.2$. The presence of other strongly contributing features ($x_2$, $x_3$ and $x_4$) in the attention module can induce complex high-order interactions, where negative interactions reduce the positive contributions of $x_1$ when it joins certain coalitions. For more detailed interaction values, please refer to Appendix C.

## 5 INTERACTION ANALYSIS

**Experimental Setup.** We conduct interaction analyses by performing experiments across practical real datasets with DNNs. We use VGG (Yan et al., 2015) and ViT (Dosovitskiy et al., 2020) models for image classification on ImageNet (Deng et al., 2009) and COCO (Lin et al., 2014) datasets. All images are divided into 64 equal-sized segments, which are used as features for Shapley interaction calculations. We perform additional experiments for natural language processing using BERT (Devlin et al., 2019) for sentiment classification on IMDB (Maas et al., 2011) dataset and report the results in the appendix.

### 5.1 INTERACTION ANALYSIS OF DEEP NEURAL NETWORKS

In this section, we demonstrate the identification of important higher-order interactions and their interpretation. Figure 2 presents two sample cases from the ViT model. We select two coalitions $R_1$ and $R_2$ with $|R| = 4$ from each input image. The coalition $R_1$ corresponds to regions relevant to the model's prediction, while $R_2$ corresponds to relatively less relevant regions. For each coalition, we sample 4th-order interaction effects from 50 random permutations, and compare the true marginal effect $v(R \cup T) - v(T)$ with the expectation and variance of sampled interactions $\Delta_R v(T)$.

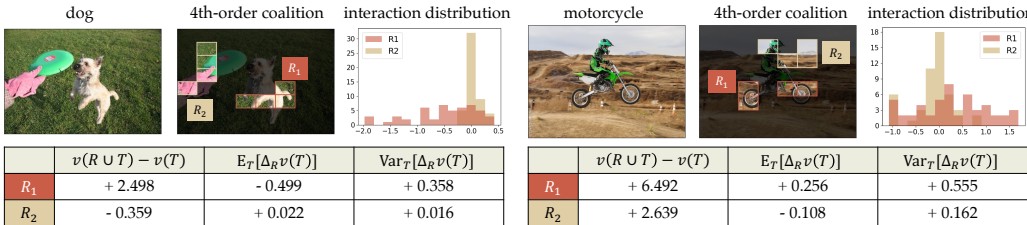

Figure 2: The distribution of interaction effects. Marginal change caused by removing $(v(R \cup T) - v(T)$ can vary significantly from the expected marginal effect $\mathbb{E}_T[\Delta_R v(T)]$ if the variance of interactions $\mathrm{Var}_T[\Delta_R v(T)]$ is large.

In both examples, $R_1$ exhibits a larger marginal effect than $R_2$, which is consistent with its stronger relevance to the model's prediction. However, the expected interaction of $R_1$ is often similar to, or even smaller than, that of $R_2$, even taking negative values. This phenomenon may be attributed to the redundancy effects among features (Fujimoto et al., 2006; Chang et al., 2025). Moreover, $R_1$ shows consistently higher variance of interactions compared to $R_2$. As discussed in Section 3.2, $\Delta_R v(T)$, the marginal effect of coalition $R$. subsumes higher-order interactions between $R$ and subsets of $T$. When the interactions vary substantially across $T$, it indicates the presence of critical higher-order interactions that influences model prediction. This discrepancy highlights the importance of accounting for higher-order interactions to obtain a more complete picture of model behavior, since expected effects may ignore critical synergistic coalitions.

## 5.2 VARIANCE AS INDICATOR OF HIGHER-ORDER INTERACTIONS

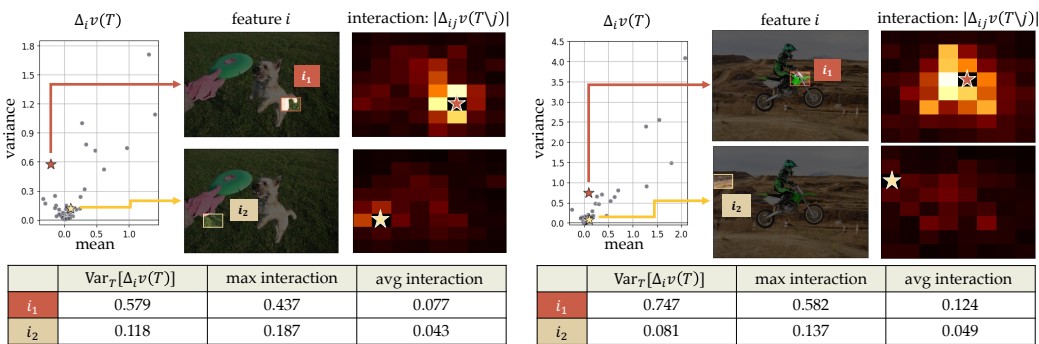

Figure 3: The relationship between marginal effect variance and interaction. Higher variance ($i_1$) is associated with greater interaction magnitude, and lower variance ($i_2$) with lower interactions.

Searching for significant higher-order interactions in high-dimensional inputs is computationally prohibitive due to the combinatorial number of possible coalitions. This intractability motivates the need for a guideline for identifying coalitions worth investigating. We find empirically that the variance of sampled contributions acts as a simple yet effective criterion. Low variance suggests that a feature does not exhibit unique interactions with other coalitions and thus does not require higher-order analysis, whereas large variance indicates the presence of critical interactions.

In Figure 3, we demonstrate this insight using examples from the ViT model. We plot the variance and expectation of $\Delta_i v(T)$ for all $i \in N$. We then select features according to their variance and examine their pairwise interactions with the other features (averaged over permutations). The results show that $i_1$ (high variance) exhibits substantially stronger interactions—both in terms of maximum and average values—than $i_2$ (low variance). These findings suggest that features with large variance of sampled contributions are promising candidates for targeted higher-order interaction analysis. Consistent with this interpretation, $i_1$ in each image corresponds to one of the main components of the object driving the model's decision (dog and motorcycle).

# 6 APPLICATIONS

Our analyses in Section 5 show that the variance of (interaction) effects provides a reliable structural signal of higher-order interactions. Motivated by this observation, we introduce the **High-Variance Effect (HIVE)** framework, a principled strategy for discovering meaningful higher-order interaction coalitions while avoiding unnecessary exploration. The HIVE framework begins by applying variance-based filtering to individual features to identify those whose effects fluctuate strongly across contexts. As shown in our analysis, such high-variance features are closely tied to the model's decision and form natural anchors for exploring higher-order interactions. Building on this, we iteratively extend the same variance-based criterion to larger coalitions: at each step, we compute the variance of (interaction) effects for candidate subsets, partition them into high-variance and low-variance groups, and expand only the supersets derived from the high-variance group. This procedure progressively focuses the search on coalitions that are most likely to carry substantial higher-order contributions, while pruning low-variance candidates that are unlikely to exhibit meaningful interactions.

In this section, we present two sets of applications. First, we evaluate whether the features identified by HIVE align with the image regions that are truly relevant to the model's decision. Second, we apply the HIVE framework iteratively to uncover higher-order synergistic coalitions. For image-based tasks, we use SLIC superpixels (Achanta et al., 2012) as feature segments. Additional quantitative analyses supporting our variance-based exploration are provided in Appendix D. Experiments on language models follow the similar procedure and are reported in Appendix E.

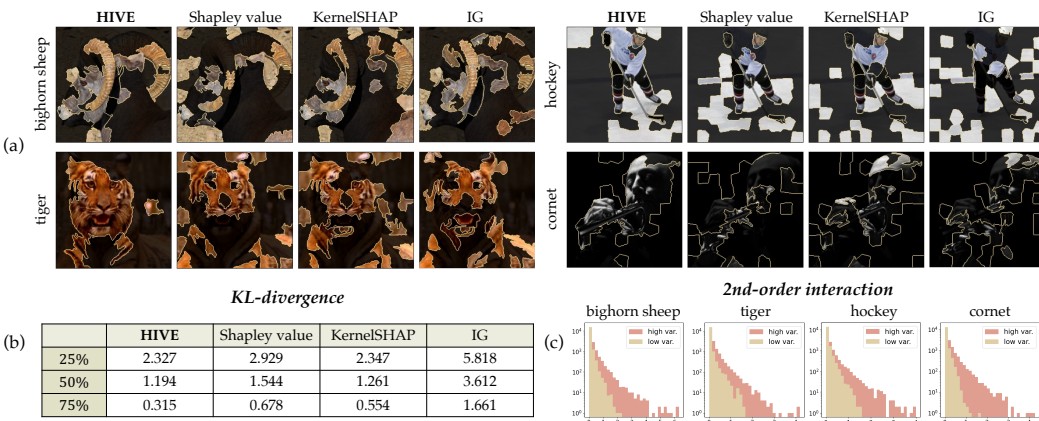

Figure 4: Regions with high variance of marginal contributions. (a) Generally, regions with higher variances are associated with segments including the main object of the class, providing a more complete understanding of important segments. (b) Inserting back top $K\%$ of segments in terms of interaction term variance causes the greatest decrease in KL-divergence from the original predictions. (c) Higher variance features have fatter-tailed distributions, i.e., many more critical interactions.

In Figure 4 (a), we compare the individual features selected by the HIVE filtering procedure with the highly attributed features identified by other attribution methods: Shapley value, KernelSHAP, and Integrated Gradients (IG), all of which follow the standard Shapley axioms (Shapley, 1953; Lundberg & Lee, 2017b; Sundararajan et al., 2017). Although variance measures the instability of contributions rather than their absolute magnitude, the highlighted regions captured by HIVE are more object-centric and thus more informative for interpretation. This aligns with the common observation that deep models for image classification rely on groups of features to represent evidential patterns. We also verify that this information can help reconstruct the model's original decision. In Figure 4 (b), we report the KL-divergence between the original logit output and the output obtained by inserting the top 25%, 50%, and 75% of segments to a blank image across 100 random samples. Variance-based selection effectively approximates the model's decision, achieving much lower average KL-divergence at all three levels compared to the other baselines.

The histograms in Figure 4 (c) show the distribution of 2nd-order interactions for top and bottom 25% segments in terms of variance for each example in Figure 4 (a). The top 25% has much fatter

tail, indicating that there are far more significant interaction terms compared to the bottom 25%. In other words, if a coalition exhibits low variance in its interactions, it likely has no substantial synergy with other features; if the variance is high, the coalition becomes a promising candidate for further exploration.

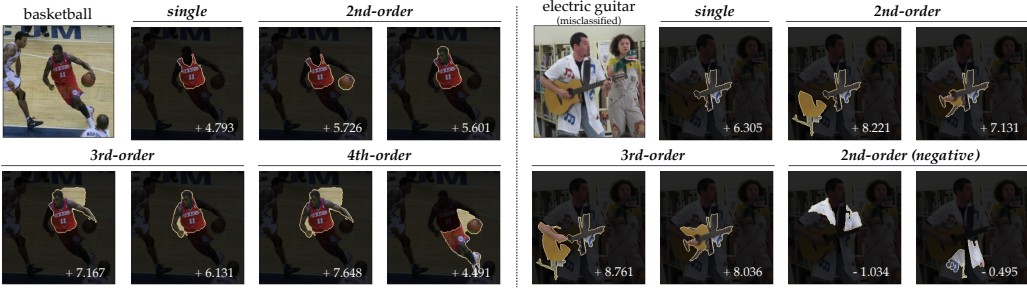

Figure 5: Examples of iterative search of critical high-order coalitions. In both the correctly classified (left) and misclassified (right) examples, critical higher-order coalitions generally include the main object of the prediction (the ball and the player in the correct classification, the guitar in the misclassification).

We apply the HIVE framework iteratively to identify higher-order synergistic coalitions, as shown in Figure 5. Given a collection $\mathcal{R}$ of candidate sets of order $k$, we compute the variance for each $R \in \mathcal{R}$ vis permutation sampling, and determine a high-variance group and a low-variance group. We then construct the candidate family for order $(k + 1)$ by expanding only the subsets in the high-variance group, while excluding those in the low-variance group from further consideration.

In Figure 5, we present one correctly classified (left) and misclassified (right) examples of discovered coalitions $R$ with their expected marginal contributions $\mathbb{E}_T[v(R \cup T) - v(T)]$ annotated at the corners of each image. Our method effectively identifies such high-order coalitions despite their sparsity. In the misclassified example, the model predicts 'electric guitar' instead of the true label 'library'. The detected higher-order coalitions are concentrated around the guitar, revealing the model's reliance on misleading evidence. Some coalitions also appear on the shirt, but their negative contributions indicate that these regions counteract the model decision instead. These results demonstrate that variance-guided exploration can effectively uncover critical higher-order coalitions and provide actionable insights into the model's decision-making.

## 7 CONCLUSION

We revisited the Shapley value by making its underlying interaction effects explicit, showing that it can be understood as decomposing the characteristic function into higher-order interaction terms and distributing each term equally among the players. This perspective extends the conventional interpretation of Shapley values as expected marginal contributions and clarifies how higher-order interactions are implicitly aggregated within them. Because this aggregation operates through expectation, it can suppress or distort meaningful higher-order interaction effects when those interactions fluctuate strongly across contexts. It leads to a structural limitation that has remained largely implicit in prior Shapley-based interpretations. Through theoretical case studies and empirical evaluations on deep neural networks, we demonstrated that the variance of low-order interaction effects reliably signals the presence of context-sensitive higher-order structure, providing a principled criterion for determining when such interactions should be explored. Building on this insight, our High-Variance Effect (HIVE) framework utilizes variance as a guidance signal to explore meaningful higher-order coalitions while pruning uninformative ones. We expect that this interaction-based perspective will advance the understanding of Shapley values and underscore the importance of explicitly analyzing higher-order interactions to obtain faithful and informative explanations of complex models.

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

# APPENDIX

## A  PROOF

**Notation.** We summarize the notations used in this paper. For convenience, we follow the simplified notations in Grabisch & Roubens (1999); Fujimoto et al. (2006). For singletons, we omit braces and write $v(i), T \cup i, T \setminus i$ instead of $v(\{i\}), T \cup \{i\}, T \setminus \{i\}$. Similarly, for multiple elements, we use $ij, ijk$ instead of $\{i, j\}, \{i, j, k\}$ when it is clear. The cardinalities of subsets $S, T, R \cdots$ are typically denoted by the corresponding lowercase letters $s, t, r, \cdots$. Moreover, $[\pi]_t$ is the subset of players up to the $t$-th player in a random permutation $\pi$, where $[\pi]_0 := \emptyset$ and $[\pi]_s := S$. $\pi^R$ denotes the set of players in $\pi$ that precede all players in $R$.

**Lemma 1.** *(k-th order interaction in $\Delta_i v(S)$) For a permutation $\pi \in \Pi(S)$, for any $t \in [0, s - k + 1]$,*

$$\Delta_i v(S) = \sum_{r=0}^{k-2} \sum_{\substack{R \subseteq S \setminus [\pi]_t \\ |R|=r}} \Delta_{i \cup R} v([\pi]_t) + \sum_{\substack{R \subseteq S \setminus [\pi]_t \\ |R|=k-1}} \Delta_{i \cup R} v(\pi^R). \tag{14}$$

*Proof.* We prove the theorem by mathematical induction on $k$. For $k = 2$, the statement holds as follows:

$$\begin{aligned}
\Delta_i v(S) &= \Delta_i v([\pi]_s) \\
&= \Delta_i v([\pi]_{s-1}) + \{\Delta_i v([\pi]_s) - \Delta_i v([\pi]_{s-1})\} \\
&= \Delta_i v([\pi]_{s-1}) + \Delta_{i \cup \pi_s} v([\pi]_{s-1}) \\
&= \cdots \\
&= \Delta_i v([\pi]_t) + \sum_{l=t}^{s} \Delta_{i \cup \pi_l} v([\pi]_{l-1}) \\
&= \Delta_i v([\pi]_t) + \sum_{j \in S \setminus [\pi]_t} \Delta_{i \cup j} v(\pi^j)
\end{aligned} \tag{15}$$

Assuming the statement holds for an integer $k = a \geq 2$, we now show that it holds for $k = a + 1$. For $k = a$, the second term becomes

$$\begin{aligned}
\sum_{\substack{R \subseteq S \setminus [\pi]_t \\ |R|=a-1}} \Delta_{i \cup R} v(\pi^R) &= \sum_{\substack{R \subseteq S \setminus [\pi]_t \\ |R|=a-1}} \left[ \Delta_{i \cup R} v([\pi]_t) + \sum_{p \in \pi^R \setminus [\pi]_t} \Delta_{i \cup R \cup p} v(\pi^p) \right] \\
&= \sum_{\substack{R \subseteq S \setminus [\pi]_t \\ |R|=a-1}} \Delta_{i \cup R} v([\pi]_t) + \sum_{\substack{R' \subseteq S \setminus [\pi]_t \\ |R'|=a}} \Delta_{i \cup R'} v(\pi^{R'}) \quad (R' = R \cup p)
\end{aligned} \tag{16}$$

Combining Equation (16) with the first term proves the statement for $k = a + 1$.

$$\Delta_i v(S) = \sum_{r=0}^{a-1} \sum_{\substack{R \subseteq S \setminus [\pi]_t \\ |R|=r}} \Delta_{i \cup R} v([\pi]_t) + \sum_{\substack{R \subseteq S \setminus [\pi]_t \\ |R|=a}} \Delta_{i \cup R} v(\pi^R). \tag{17}$$

$\square$

**Theorem 1.** *(k-th order interaction representation of a set function) A set function $v : 2^N \to \mathbb{R}$ can be expressed with k-th order interaction terms:*

$$v(N) = \sum_{r=0}^{k-1} \sum_{\substack{R \subseteq N \\ |R|=r}} \Delta_R v(\emptyset) + \sum_{\substack{R \subseteq N \\ |R|=k}} \sum_{T \subseteq N \setminus R} \frac{k}{n} \binom{n-1}{t}^{-1} \Delta_R v(T) \tag{18}$$

*Proof.* For any permutation $\pi \in \Pi(N)$, the following equation holds:

$$v(N) = \sum_{t=0}^{n-1} \Delta_{\pi_{t+1}} v([\pi]_t). \tag{19}$$

Apply Lemma 1 to the expectation of these forms with random permutations. Then, $v(N)$ is represented as the weighted sum of $\Delta_R v(\emptyset)$ with $|R| \in [1, k-1]$ and $\Delta_R v(T)$ with $|R| = k, T \subseteq N \setminus R$.

For $|R| \in [1, k-1]$, $\Delta_R v(\emptyset)$ appears in all permutations. So the weight is 1.

For $|R| = k$, count the number of appearance of $\Delta_R v(T)$.

$$\frac{1}{n!} \cdot t! \cdot k \cdot (n - t - 1)!$$
$$= \frac{k}{n} \binom{n-1}{t}^{-1} \tag{20}$$

Using Theorem 2 and the efficiency property of the Shapley value, the same result can be easily obtained.

$\square$

**Theorem 2.** *(k-th order interaction representation) The Shapley value can be represented in terms of k-th order interactions:*

$$\phi_i(v) = \sum_{r=0}^{k-2} \frac{1}{r+1} \sum_{\substack{R \subseteq N \setminus i \\ |R| = r}} \Delta_{i \cup R} v(\emptyset) + \sum_{t=0}^{n-k} \frac{1}{n} \binom{n-1}{t}^{-1} \sum_{\substack{R \subseteq N \setminus i \\ |R| = k-1}} \sum_{\substack{T \subseteq N \setminus (i \cup R) \\ |T| = t}} \Delta_{i \cup R} v(T). \tag{21}$$

*Proof.* Note that the Shapley value is represented as follows:

$$\frac{1}{n!} \sum_{\pi \in \Pi(N)} \Delta_i v(\pi^i). \tag{22}$$

Apply this representation for the second term in Lemma 1. Check the weight of $\Delta_{i \cup R} v(T)$ for a given $R, T$, by counting the number of appearance in all permutations. $p$ denotes the index of $i$ in the given permutation.

$$\frac{1}{n!} \cdot t! \cdot \sum_{p=k+t}^{n} \binom{p-t-2}{k-2} \cdot (k-1)! \cdot (n-k-t)!$$
$$= \frac{t!}{n!} \cdot \binom{n-t-1}{k-1} \cdot (k-1)! \cdot (n-k-t)! \tag{23}$$
$$= \frac{1}{n} \binom{n-1}{t}^{-1}$$

$\square$

**Theorem 3.** *(permutation sampling) The Shapley value with k-th order interaction can be estimated through permutation sampling:*

$$\phi_i(v) = \sum_{\substack{R \subseteq N \setminus i \\ |R| \in [0, k-2]}} \frac{1}{r+1} \Delta_{i \cup R} v(\emptyset) + \frac{1}{k-1} \sum_{t=0}^{n-k} \mathbb{E}_{\pi \in \Pi(N)} \left[ \sum_{\substack{R \subseteq N \setminus [\pi]_{t+1} \\ |R| = k-2}} \Delta_{i \pi_{t+1} \cup R} v([\pi]_t) \right] \tag{24}$$

*Proof.* Check the coefficient of $\Delta_{i \cup R} v(T)$ by calculating the expectation part.

$$\frac{1}{n!} \cdot t! \cdot (k-1) \cdot (n-t-1)!$$
$$= (k-1) \cdot \frac{1}{n} \binom{n-1}{t}^{-1} \tag{25}$$

Then, the second term is the same as the second term in Theorem 2. $\square$

**Theorem 4.** *(dividends in $k$-th order interaction representation) The Harsanyi dividend of $L \subseteq N$ is embedded in the $k$-th order interaction representation of Shapley value as follows:*

$$\phi_i(v) = \sum_{r=0}^{k-2} \frac{1}{r+1} \sum_{\substack{R \subseteq N \setminus i \\ |R|=r}} \alpha_{i \cup R} + \sum_{\substack{R \subseteq N \setminus i \\ |R|=k-1}} \sum_{\substack{L \subseteq N \\ (i \cup R) \subseteq L}} \frac{1}{k} \binom{l}{l-k}^{-1} \alpha_L. \tag{26}$$

*Proof.* Let $l' := l - k$. $\alpha_L$ appears in $\Delta_{i \cup R} v(T)$ when $T$ includes $L \setminus (i \cup R)$. Therefore, for given $R, L$, count the number of permutations where $L \setminus (i \cup R) \subseteq \pi^R$ in $\Delta_i v(\pi^i)$ as done in the proof of Theorem 2. $t$ denotes the size of $\pi^R$.

$$\frac{l'!}{n!} \cdot \sum_{t=l'}^{n-k} \binom{t}{l'} \sum_{p=k+t}^{n} \binom{p-t-2}{k-2} \cdot (k-1)! \cdot (n-k-l')!$$

$$\frac{l'!}{n!} \cdot (k-1)! \cdot (n-k-l')! \cdot \sum_{t=l'}^{n-k} \binom{t}{l'} \binom{n-t-1}{k-1} \tag{27}$$

Using Vandermonde's identity, we obtain

$$\frac{l'!}{n!} \cdot (k-1)! \cdot (n-k-l')! \cdot \binom{n}{l'+k}$$

$$= \frac{1}{k} \binom{l'+k}{l'}^{-1} \tag{28}$$

$$= \frac{1}{k} \binom{l}{l-k}^{-1}$$

$\square$

## B  OTHER RELATED WORK

**Game-theoretical model interpretation.** Modern model interpretation methods aim to explain complex models by quantifying the contribution of each input feature to the model's output (Ribeiro et al., 2016; Lundberg & Lee, 2017b; Sundararajan et al., 2017). A variety of feature attribution techniques have been proposed, including gradient-based and perturbation-based approaches (Binder et al., 2016; Zhou et al., 2016; Smilkov et al., 2017; Selvaraju et al., 2017; Montavon et al., 2017; Shrikumar et al., 2017; Nam et al., 2020; Kapishnikov et al., 2021). Although effective in practice, these methods are largely heuristic and lack rigorous theoretical guarantees. In contrast, game-theoretical techniques approach model prediction as a cooperative game, where features act as players contributing to the overall payoff (Rozemberczki et al., 2022). Among these methods, Shapley value–based attributions are axiomatically grounded: they uniquely and fairly distribute the model prediction among input features by satisfying four axioms—efficiency, symmetry, dummy, and additivity (Sundararajan & Najmi, 2020).

**Interactions.** Feature interaction refers to the additional contribution that arises when a set of players act together beyond their individual effects (Grabisch & Roubens, 1999). This notion can be formalized through the Harsanyi dividend, which decomposes any cooperative game into context-free coalition contributions (Harsanyi, 1982; Fujimoto et al., 2006). Since each dividend represents the pure interaction of a coalition, the Shapley value can be seen as the sum of all interaction terms involving a given player (Grabisch & Roubens, 1999; Dehez, 2017). This insight is especially important for highly nonlinear models, where interactions can dominate the predictions and must be explicitly treated to obtain faithful and robust explanations (Singhvi et al., 2024). However, because the Shapley value provides only an additive allocation, it conflates main effects with interaction effects rather than disentangling them. Chang et al. (2025) partially address this gap by reformulating the Shapley value as a weighted sum of 2nd-order interactions via permutation sampling.

## C INTERACTIONS IN CASE STUDY EXAMPLES

We describe the interaction values for the case study examples in Section 4. Tables 1 and 2 report the detailed marginal contribution and interaction values of $x_5$ in the max function and $x_1$ in the attention module example. The results show that the marginal contributions of $x_5$ and $x_1$ vary drastically depending on $T$. In particular, the contribution decreases—and even becomes negative—when other players are present. Similar variations are observed in higher-order interactions, where the signs of 2nd-, 3rd-, and 4th-order terms fluctuate substantially across coalitions. These results indicate that variables participating with other strongly contributing features in the max function and attention module can exhibit complex interaction structures, including frequent negative interactions. In such cases, the expectation-based computation of Shapley values may obscure the positive contributions of certain players in specific coalitions, highlighting the need to explicitly analyze higher-order interactions to capture their detailed effects.

Table 1: Marginal contribution and interaction values $\Delta_R v(T)$ for the max function example, focusing on player $x_5$. Columns correspond to the context coalitions $T$ and rows to $R$.

| $R \backslash T$ | $\emptyset$ | $\{2\}$ | $\{3\}$ | $\{4\}$ | $\{2,3\}$ | $\{3,4\}$ | $\{2,4\}$ | $\{2,3,4\}$ |
|---|---|---|---|---|---|---|---|---|
| $\{5\}$ | 10.0 | 3.0 | 2.0 | 1.0 | 2.0 | 1.0 | 1.0 | 1.0 |
| $\{5,2\}$ | -7.0 | - | 0.0 | 0.0 | - | 0.0 | - | - |
| $\{5,3\}$ | -8.0 | -1.0 | - | 0.0 | - | - | 0.0 | - |
| $\{5,4\}$ | -9.0 | -2.0 | -1.0 | - | -1.0 | - | - | - |
| $\{5,2,3\}$ | 7.0 | - | - | 0.0 | - | - | - | - |
| $\{5,2,4\}$ | 7.0 | - | 0.0 | - | - | - | - | - |
| $\{5,3,4\}$ | 8.0 | 1.0 | - | - | - | - | - | - |
| $\{5,2,3,4\}$ | -7.0 | - | - | - | - | - | - | - |

Table 2: Marginal contribution and interaction values $\Delta_R v(T)$ for the attention module example, focusing on player $x_1$. Columns correspond to the context coalitions $T$ and rows to $R$.

| $R \backslash T$ | $\emptyset$ | $\{2\}$ | $\{3\}$ | $\{4\}$ | $\{2,3\}$ | $\{3,4\}$ | $\{2,4\}$ | $\{2,3,4\}$ |
|---|---|---|---|---|---|---|---|---|
| $\{1\}$ | 1.4261 | 0.5474 | 0.1080 | -0.0018 | -0.0697 | -0.4128 | -0.1383 | -0.3761 |
| $\{1,2\}$ | -0.8787 | - | -0.1778 | -0.1365 | - | 0.0366 | - | - |
| $\{1,3\}$ | -1.3181 | -0.6171 | - | -0.4110 | - | - | -0.2378 | - |
| $\{1,4\}$ | -1.4279 | -0.6857 | -0.5208 | - | -0.3064 | - | - | - |
| $\{1,2,3\}$ | 0.7009 | - | - | 0.1732 | - | - | - | - |
| $\{1,2,4\}$ | 0.7422 | - | 0.2144 | - | - | - | - | - |
| $\{1,3,4\}$ | 0.9071 | 0.3793 | - | - | - | - | - | - |
| $\{1,2,3,4\}$ | -0.5278 | - | - | - | - | - | - | - |

# D    JUSTIFICATION OF THE VARIANCE-BASED EXPANSION STRATEGY

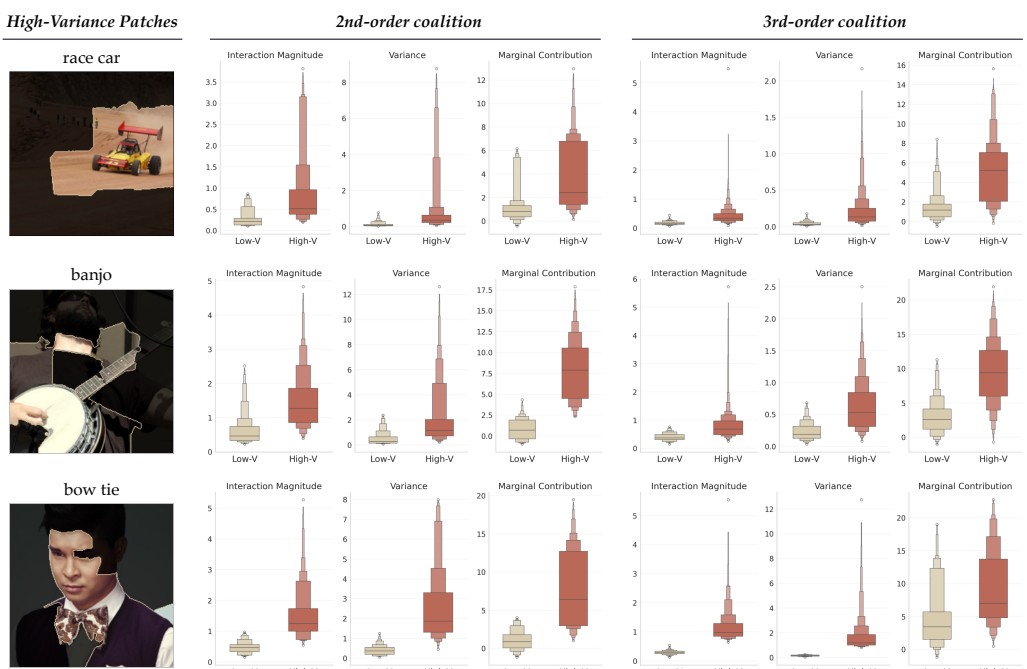

Figure 6: Comparison between supersets derived from high-variance vs. low-variance feature groups. High-variance supersets consistently yield stronger and more context-dependent interaction terms, validating variance as an effective criterion for guiding higher-order coalition exploration.

To justify the use of variance as a criterion for guiding the exploration of higher-order coalitions, we conduct a quantitative analysis using a VGG-based ImageNet classifier with 15 SLIC segments per image. For each first-order feature, we sample $\Delta_i v(T)$ across sampled permutations and compute its variance. As shown in Figure D, high-variance features spatially align with the regions most responsible for the model's prediction, indicating that variance captures semantically meaningful and influential feature behavior.

We then evaluate whether high-variance features indeed serve as better building blocks for constructing higher-order coalitions. Specifically, we construct two groups: high-variance (top 30%) and low-variance (bottom 30%). From these groups, we construct two families of supersets at interaction order 2: (1) Low-V supersets, which include at least one low-variance feature but exclude all high-variance features; (2) High-V supersets, which include at least one high-variance feature but exclude all low-variance features. The same grouping procedure is applied to second-order feature coalitions to construct candidate third-order coalitions.

For each coalition $R$ in these families, we measure the interaction magnitude $\mathbb{E}_T[\|\Delta_R v(T)\|]$, the variance $\text{Var}_T[\|\Delta_R v(T)\|]$, and the expected marginal contribution ($\mathbb{E}_T[v(R \cup T) - v(T)]$). The expected marginal contribution indicates how strongly the coalition impacts the network's inference.

In Figure D, our results show a clear pattern. Low-V supersets consistently exhibit small interaction magnitude and low interaction variance, indicating that they do not meaningfully participate in higher-order effects. Consequently, extending such coalitions provides little benefit and would only increase computational overhead. This demonstrates that variance serves as an effective pruning criterion, substantially reducing the number of evaluations for higher-order interactions.

Their low expected marginal contributions further confirm that these coalitions have minimal direct influence on the model's inference. Consequently, extending such coalitions provides little benefit and would only increase computational overhead. This demonstrates that variance serves as an effective pruning mechanism, substantially reducing the number of higher-order coalitions that must be explored.

Conversely, High-V supersets display substantially larger interaction magnitudes, higher interaction variance, and notably larger marginal contributions. These characteristics indicate the presence of non-negligible and context-dependent higher-order structure. Such supersets are precisely the coalitions whose interactions cannot be reliably summarized via expectations and therefore should be prioritized for higher-order evaluation. Moreover, the large marginal contributions imply that these coalitions directly influence the model's decision-making process, validating that our variance-guided expansion strategy not only avoids unnecessary exploration but also directs computation toward the most inference-critical feature combinations.

| | 2nd-order coalition | | | | | | | 3rd-order coalition | | | | | |
|---|---|---|---|---|---|---|---|---|---|---|---|---|---|
| race car | Marginal Contribution | Variance | SII | BII | STI | FaithShap | | Marginal Contribution | Variance | SII | BII | STI | FaithShap |
| High-V | + 3.845 | 0.916 | + 0.034 | + 0.177 | - 0.105 | + 0.103 | | + 4.975 | 0.216 | + 0.082 | + 0.016 | + 0.176 | + 0.035 |
| Low-V | + 1.216 | 0.113 | + 0.015 | - 0.006 | - 0.019 | + 0.012 | | + 1.454 | 0.041 | + 0.004 | + 0.004 | + 0.013 | + 0.004 |

| | | | | | | | | | | | | | |
|---|---|---|---|---|---|---|---|---|---|---|---|---|---|
| banjo | Marginal Contribution | Variance | SII | BII | STI | FaithShap | | Marginal Contribution | Variance | SII | BII | STI | FaithShap |
| High-V | + 7.934 | 2.161 | - 0.349 | - 0.267 | - 0.415 | - 0.305 | | + 9.394 | 0.625 | + 0.065 | - 0.009 | + 0.174 | + 0.010 |
| Low-V | + 0.768 | 0.536 | - 0.057 | - 0.010 | - 0.122 | - 0.028 | | + 2.951 | 0.231 | + 0.043 | + 0.002 | + 0.104 | + 0.008 |

| | | | | | | | | | | | | | |
|---|---|---|---|---|---|---|---|---|---|---|---|---|---|
| bow tie | Marginal Contribution | Variance | SII | BII | STI | FaithShap | | Marginal Contribution | Variance | SII | BII | STI | FaithShap |
| High-V | + 7.583 | 2.562 | - 0.217 | + 0.260 | - 0.326 | + 0.010 | | + 8.537 | 1.189 | + 0.179 | - 0.032 | + 0.593 | + 0.029 |
| Low-V | + 1.045 | 0.400 | - 0.001 | + 0.124 | - 0.110 | + 0.068 | | + 4.252 | 0.246 | + 0.084 | + 0.014 | + 0.216 | + 0.026 |

Figure 7: Comparison with existing interaction indices. Expectation-based interaction indices (SII, BII, STI, Faith-Shap) fail to distinguish the two superset families, whereas variance reliably separates them by capturing context-sensitive higher-order effects that expectation-based measures overlook.

To demonstrate that variance provides information beyond existing interaction indices, we compare several widely used Cardinal Interaction Indices: the Shapley Interaction Index (SII), the Banzhaf Interaction Index (BII) (Grabisch & Roubens, 1999), the Shapley–Taylor Interaction Index (STI) (Sundararajan et al., 2020), and Faith-Shap (Tsai et al., 2023). All of these methods evaluate interactions through an expectation (or weighted summation) over discrete derivatives, and therefore belong to the Cardinal Interaction Index (CII) class.

Our aim is to assess whether these indices can distinguish two families of supersets: those derived from high-variance coalitions versus those from low-variance coalitions. Figure 7 reports the average values for each family. All existing indices produce similarly small values for both groups and therefore fail to separate them, even though the two families differ markedly in their true marginal contributions and interaction variance.

This limitation stems from the fact that expectation-based indices inherit the sign-cancellation problem of discrete derivatives in deep neural networks. When positive and negative interactions oscillate across contexts, their aggregated value collapses toward zero. In contrast, variance does not suffer from this cancellation and provides diagnostic information that reveals which coalitions exhibit meaningful, context-sensitive higher-order interactions. As a result, variance serves as a far more reliable criterion for guiding higher-order coalition expansion than existing interaction indices.

# E  INTERACTION ANALYSIS IN LANGUAGE MODELS

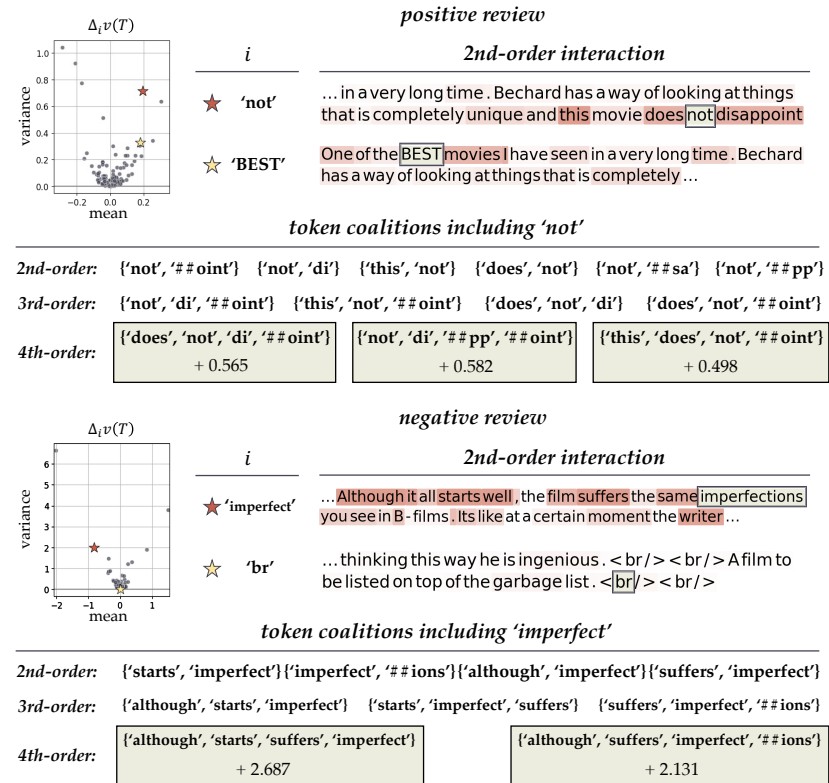

Figure 8: Interaction analysis on language model. For each case, a token with high variance of marginal contributions is shown to actively interact with other tokens. By examining higher-order coalitions involving this token, we identify meaningful token coalitions that play a decisive role in the model's prediction.

We conduct additional experiments in natural language processing to examine interaction effects in language inference tasks. Specifically, we use a BERT-based sentiment classifier (Devlin et al., 2019) trained on the IMDB dataset (Maas et al., 2011), which predicts whether a given movie review is positive or negative. We analyze two representative samples—one positive and one negative review—by sampling marginal contributions for individual tokens and computing their mean and variance.

In the positive example, the tokens 'not' and 'BEST' have similar expected contributions but very different variances. 'BEST' consistently supports the positive prediction by increasing the logit output regardless of the presence of other tokens. On the other hand, the token 'not' by itself indicates negativity; however, we observe that tokens in 'this movie does not disappoint' have a large magnitude of interaction values with 'not'. Following the approach in Section 6, we investigate higher-order coalition structures by focusing on low-order coalitions with high variance. The token 'not' frequently forms coalitions with 'disappoint', and these coalitions yield substantially larger marginal contributions than the expected contribution of the single token 'not'.

In the negative case, tokens without semantic meaning, such as 'br', '/', and '<', generally have the lowest variance of contributions, which implies that they do not interact with any other tokens to construct sentence context. However, the token 'imperfect' has a smaller expected contribution than 'br' and much higher variance. This token commonly interacts with 'suffer', forming a coalition that conveys the reviewer's dissatisfaction. Such coalitions have significantly larger marginal contributions, thereby driving the negative classification.

# F  INTERACTION ESTIMATION ACCURACY

Figure 9: Interaction Estimation Accuracy: Set-based vs. Permutation-based Estimation.

We compare the estimation accuracy of the set-based estimator (Theorem 2) and the permutation-based estimator (Theorem 3) using a VGG network trained on ImageNet. Each image is partitioned into 15 segments using SLIC. To compute ground-truth interaction values, we evaluate the exact interaction terms for selected feature subsets and measure the absolute error between these ground-truth values and their corresponding estimates. Since most interactions are near zero due to sparsity, we first identify non-negligible interactions using our variance-based filtering and compute estimation errors only on these informative subsets that are constructed from features with high-variance effects.

Figure 9 reports the estimation error across the number of sampled sets/permutations, evaluated over five examples. Each subplot corresponds to a different interaction order. The y-axis shows the sum of estimation errors over 30 randomly selected subsets (log-scale). The results indicate that the set-based estimator is substantially more sensitive to the particular sampled context sets, leading to slower and less stable convergence. In contrast, the permutation-based estimator converges more smoothly and consistently to the ground-truth interaction values. Beyond stability, it also achieves significantly lower estimation errors for the same number of evaluations, demonstrating that permutation-based sampling provides a more efficient and reliable strategy for estimating higher-order interaction terms.

