# OpenReview forum: "Analysis of High-order Interactions in Shapley value for Model Interpretation"
_ICLR.cc/2026/Conference — Submitted to ICLR 2026_

### Official Review · Reviewer_HZ32 · 2025-10-23

**Soundness:** 4
**Presentation:** 2
**Contribution:** 2
**Rating:** 4
**Confidence:** 4

**Summary:**

This paper proposes an interaction representation of the Shapley value and the associated cooperative game $\nu$. This interaction representation relies on discrete derivatives $\Delta_S\nu(T)$ with $\vert S \vert \leq k$ for a chosen maximum interaction order $k$. The representation is an extension of the second-order ($k=2$) representation for the Shapley value. (Chang et al., 2025) to higher order $k>2$. The paper then observes that this representation can be viewed as an expectation over random permutations, similar to permutation sampling for the Shapley value. For each sampled permutation, the corresponding discrete derivatives are computed and then averaged. Conequently, within the computation, discrete derivatives are evaluated for multiple $T$, where the variance between these values is viewed as an indicator for higher-order interactions, e.g. the variance of the marginal contributions $\mathbb{V}_T[\Delta_i\nu(T)]$ is an indicator for interactions, since it quantifies the non-additivity of the game.
The authors then validate their approach empirically on synthetic functions, and investigate interactions in deep neural networks.

**Strengths:**

- The paper is well structured and the theoretical results seem sound
- Analyzing higher-order interactions to understand model decision is very important
- Decomposing the Shapley value into higher-order interactions is an interesting approach that extends the initial approach by Chang et al. (2025), and provides a useful tool for analysis.
- Investigating the variance of discrete derivatives / marginal contributions as a measure of interaction is an interesting and novel idea
- Efficient computation of Shapley interactions via extended permutation sampling is an interesting research direction.
- Analyzing average discrete derivatives as a measure of interaction is a well-known and useful technique

**Weaknesses:**

- This paper is very poorly embedded in the current feature interaction literature, where several important papers were not discussed:
  - The Shapley interaction index as the first measure of interaction in cooperative games extends the Shapley value to higher-order interactions. More importantly, the Shapley interaction index is an average of discrete derivatives (over T), very related to the second sum in Eq. (10) (see questions). Moreover, the Shapley interaction index also satisfies a recursive property that relates higher-order interactions to lower-order interactions, e.g. the Shapley value.
  - All interaction indices [1-5] can be viewed as average (over T) discrete derivatives, similar to the measure that appears as the second part of the sum in Eq. (10). It is crucial to understand the differences between these measures, e.g. it was shown that they all summarize higher-order Harsanyi dividends differently [Proof of Theorem 8, 5]. It seems that this measure of interaction used here could be very well related to existing interaction measures. An additional analysis in terms of the Möbius transform would be very helpful.
  - In [4] there were already two permutation sampling approaches for interaction measures proposed. In addition to the theoretical comparison of the average discrete derivatives, it seems that these approaches use similar ideas.
 - Besides sampling permutations, the method requires evaluating several discrete derivatives. An analysis of the complexity of these calculations is missing.
- Depending on the connection to existing Shapley interaction indices, the approximation method could be compared against existing methods to compute Shapley interactions [4,6]. If this interaction measure is indeed novel, then heuristics could be used to evaluate the approximation quality.

Overall, in its current state, it is difficult to judge how this approach is related to ongoing efforts of analyzing interactions. It would be great, if the authors could clarify this.

**Questions:**

1. What is the connection of the average discrete derivatives used in the second part of the sum in Eq. (10) as an interaction measure compared to:
- The Shapley interaction index [1]?
- The Shapley Taylor interaction index [2]?
- The n-Shapley values [3,5]? Specifically, the n-Shapley values are constructed such that the second-order interactions can be added to the first-order values to obtain the Shapley values.
- The Faithful Shapley interaction index [4]?


2. How does your expectation over discrete derivatives summarize the Harsanyi dividends?
3. How does your permutation sampling approach relate to the permutation sampling extensions used as baselines in [4]

**Literature**

[1] Grabisch, Michel, and Marc Roubens. "An axiomatic approach to the concept of interaction among players in cooperative games." International Journal of game theory 28.4 (1999): 547-565.

[2] Sundararajan, Mukund, Kedar Dhamdhere, and Ashish Agarwal. "The shapley taylor interaction index." International conference on machine learning. PMLR, 2020.

[3] Lundberg, Scott M., et al. "From local explanations to global understanding with explainable AI for trees." Nature machine intelligence 2.1 (2020): 56-67.

[4] Tsai, Che-Ping, Chih-Kuan Yeh, and Pradeep Ravikumar. "Faith-shap: The faithful shapley interaction index." Journal of Machine Learning Research 24.94 (2023): 1-42.

[5] Bordt, Sebastian, and Ulrike von Luxburg. "From Shapley values to generalized additive models and back." International Conference on Artificial Intelligence and Statistics. PMLR, 2023.

[6] Fumagalli, Fabian, et al. "SHAP-IQ: Unified approximation of any-order shapley interactions." Advances in Neural Information Processing Systems 36 (2023): 11515-11551.

[7] Muschalik, Maximilian, et al. "shapiq: Shapley interactions for machine learning." Advances in Neural Information Processing Systems 37 (2024): 130324-130357.

---

> ### Author Response · Authors · 2025-11-26
>
> Thank you for the thoughtful and constructive feedback. We have carefully reviewed the suggested references and integrated their perspectives into our discussion. To clarify our motivation, contributions, and relation to prior work, we have strengthened Section 3.4 with an explicit proof of the connection to Harsanyi dividends, added a new Section 3.5 (Relation to Prior Work), and included several additional experiments in the Appendix. We would be grateful if you could also consider the overall narrative presented in our general response.
>
> **Q1. The connection of the average discrete derivatives used in the second part of the sum in Eq. (10) as an interaction measure compared to Related Work.**
>
> Our representation in Theorem 2 arises from a different starting point than prior work: we begin by decomposing the marginal contribution itself into a consecutive sum of interaction effects, as formalized in Lemma 1. When this decomposition is performed at order $k$, the resulting second term naturally becomes the average of all $k$-th order discrete derivatives computed over different contexts $T$. Although derived from a distinct perspective, this term aligns with the standard characterization of interaction indices as weighted sums of discrete derivatives, as established in the classical Cardinal Interaction Index (CII) framework [4].
>
> Among the major interaction indices, STI [5], Faith-Shap [6], and n-Shapley [1] all satisfy the efficiency axiom and recover a GAM representation. When extended to the full cardinality, they result in the Harsanyi dividend representation. Because our approach explicitly decomposes marginal contributions into consecutive interaction components (from individual marginal contributions to Harsanyi dividends), the components in Theorem 2 for all $\phi_i(v)$ also satisfies the efficiency axiom by construction. This structure makes the max-order term accumulate all higher-order effects in an averaged manner, which directly connects our framework to the intuition behind the interaction-distribution axiom of STI. In particular, the second term of Theorem 2 corresponds to an aggregation of STI-style interactions that are uniformly allocated to all features in $i\cup R$.
>
> However, expectation-based interaction evaluation can obscure important behaviors, especially in non-convex models. As noted in Section 3.4, discrete derivatives may flip sign depending on the conditioning set $T$, causing higher-order interactions to cancel out and yield near-zero attributions, even when strong effects are present. Similar observations about redundancy and sign-cancellation have been reported in other literature [1,2,3]. This motivates our proposal to examine not only the expected interaction term but also its deviation across sampled coalitions (Theorem 2). This deviation explicitly reveals which coalitions introduce inconsistent interaction behavior and should be prioritized for deeper or higher-order evaluation (Analysis in Section 5). We believe this perspective is complementary to existing interaction research. While prior works focus on defining principled weighted-sum indices or efficient approximation schemes, our decomposition highlights *when and where* expectation-based measures may become unreliable, and provides a mechanism to guide the exploration of meaningful coalitions in high-order for complex models.
>
> **Q2. How does your expectation over discrete derivatives summarize the Harsanyi dividends?**
>
> The expected interaction associated with the coalition $i \cup R$ can be expressed as
>
> $$
> E_T[\Delta_{i\cup R}v(T)]=\sum_{(i \cup R)\subseteq L \subseteq N}\frac{1}{k}\,\binom{l}{l-k}^{-1}\,\alpha_L
> $$
> where $l=|L|$. It shows that our expectation corresponds to a specific weighted average of the Harsanyi dividends $\alpha_L$ over all supersets $L$ that contain $i \cup R$. This expression follows directly from the similar permutation-counting argument used in the proof of Theorem 2, together with Lemma 1: counting the number of permutations where $L\setminus(i\cup R)\subseteq \pi ^ {R}$ in $\Delta_iv(\pi^i)$. For completeness, we have included the full derivation in the Appendix of the revised manuscript.
>
> Since each $\Delta_R v(T)$ can be decomposed into higher-order Harsanyi dividends, a high variance implies that there exist subsets $S$  with non-negligible Harsanyi dividends $\alpha_{S\cup R}$, which contribute to $\Delta_R v(T)$ only when the additional features $S$ are included in the context $T$. That is, it signals the presence of context-dependent higher-order interactions that cannot be captured by the expectation alone and therefore warrant further exploration.

---

> ### Author Response · Authors · 2025-11-26
>
> **Q3. How does your permutation sampling approach relate to the permutation sampling extensions used as baselines in Faith-Shap?**
>
> The permutation-sampling baselines estimate all $k$-th order discrete derivatives for every subset, which leads to high computational cost. While each derivative evaluation in our method has a similar per-sample complexity, the overall strategy (utilized in Section 6) is different.
> Our approach expands interactions incrementally:
> 1. Start from low-order interactions,
> 2. measure their variance across permutations, and
> 3. **extend only those subsets whose variance indicates meaningful higher-order effects**.
>
> This avoids computing derivatives for the majority of $k$-order subsets, yielding substantial savings in sparse-interaction settings common in deep models.
>
> In contrast, STI first selects features using a separate importance method (SV or IG), which may miss variables with near-zero expectation but high-variance interactions. Our variance-guided expansion naturally captures such cases.
>
> The strength of our pipeline lies in **reducing the number of candidate coalitions** before higher-order evaluation. Both Figure 4 (c) in the main paper and the additional experiment in Appendix D (Figure 6 in the revised manuscript) show that **only supersets derived from high-variance coalitions produce meaningful higher-order interactions**. Low-variance coalitions consistently yield negligible interaction magnitude, indicating that they require no further expansion. Thus, the variance criterion eliminates the vast majority of supersets from consideration. This pruning effect becomes especially powerful in models with sparse interaction structure, such as large language models [7]. In such settings, meaningful interactions are concentrated in a small number of feature groups, and our variance-guided expansion reduces the combinatorial search space dramatically.
>
> [1] Bordt, Sebastian, and Ulrike von Luxburg. "From Shapley values to generalized additive models and back." *International Conference on Artificial Intelligence and Statistics*. PMLR, 2023.
>
> [2] Fujimoto, Katsushige, Ivan Kojadinovic, and Jean-Luc Marichal. "Axiomatic characterizations of probabilistic and cardinal-probabilistic interaction indices." *Games and Economic Behavior* 55.1. 2006.
>
> [3] Chang, Wonjoon, Myeongjin Lee, and Jaesik Choi. "Rethinking Shapley Value for Negative Interactions in Non-convex Games." *The Thirteenth International Conference on Learning Representations*. 2025.
>
> [4] Grabisch, Michel, and Marc Roubens. "An axiomatic approach to the concept of interaction among players in cooperative games." *International Journal of game theory* 28.4. 1999.
>
> [5] Sundararajan, Mukund, Kedar Dhamdhere, and Ashish Agarwal. "The shapley taylor interaction index." *International conference on machine learning*. PMLR, 2020.
>
> [6] Tsai, Che-Ping, Chih-Kuan Yeh, and Pradeep Ravikumar. "Faith-shap: The faithful shapley interaction index." *Journal of Machine Learning Research* 24.94. 2023.
>
> [7] Kang, Justin Singh, et al. "Spex: Scaling feature interaction explanations for llms." .In Forty-second International Conference on Machine Learning, 2025.

---

### Official Review · Reviewer_z4D4 · 2025-10-28

**Soundness:** 4
**Presentation:** 3
**Contribution:** 3
**Rating:** 2
**Confidence:** 5

**Summary:**

The paper presents a novel way of identifying higher-order interactions to summarize model behavior and improve model explanations such as feature attributions. Therein, the paper analyzes the Shapley value and its characteristic functions (value function) and proposes a novel representation of the Shapley value and the value function in terms of higher-order interactions of a fixed order $k$. The paper then uses this representation to propose an estimator for these interactions and a scheme to systematically search for interesting coalitions/interactions.

---

Overall I think the contribution of the work is **solid and worthy of acceptance**, once the problems with the related work and therefore rather weak empirical evaluation is addressed.

 ---
### References
The following references are used throughout the review
- [1] SPEX: https://openreview.net/pdf?id=UQpYmaBGwB
- [2] ProxySPEX: https://arxiv.org/abs/2505.17495
- [3] shapiq: https://proceedings.neurips.cc/paper_files/paper/2024/file/eb3a9313405e2d4175a5a3cfcd49999b-Paper-Datasets_and_Benchmarks_Track.pdf
- [4] FaithSHAP: https://jmlr.org/papers/v24/22-0202.html
- [5] Shapley Taylor: https://proceedings.mlr.press/v119/sundararajan20a/sundararajan20a.pdf
- [6] Efficient Shapley Interaction Index: https://proceedings.mlr.press/v206/bordt23a/bordt23a.pdf
- [7] SHAP-IQ: https://proceedings.neurips.cc/paper_files/paper/2023/file/264f2e10479c9370972847e96107db7f-Paper-Conference.pdf
- [8] SVARM-IQ https://proceedings.mlr.press/v238/kolpaczki24a/kolpaczki24a.pdf

**Strengths:**

- **Good Contribution:** All in all I really like the contribution of the paper! I think it's clearly presented, well based on the fundamental works and *complements* the current research stream of interaction quantification based on Shapley Interactions. However, I think this work's contribution would greatly benefit from a proper discussion and comparison of the related work, which is currently missing.
---
- **Well Written and Presented:** The writing and presentation of the paper is of a very high quality. The paper flows well and shows its results clearly. While I needed some time to wrap my head around Figure 1 and Lemma 1 the main part of the paper is very clear.
---
- **Good Case Studies:** I think the current empirical evaluation is already showing why the method is useful and can be applied to identify interesting interactions. While I think that a better comparison to the current literature on interaction quantification is needed, the quality of what's already there is quite high.

**Weaknesses:**

- **Poor Related Work:** The paper **basically disregards** the last three to four years of Shapley interaction research and does not acknowledge or reference to it. While I agree with the authors when they write *"Note that the term ‘interaction’ in this study indicates causal interaction to understand implicit interaction effects behind the Shapley value, not interaction index in game theory literature, which provides a generalized allocation framework for a subset of players."* that the contribution is different (see strengths) the related work on computing Shapley Interactions still cannot be neglected. The following contains a non-exhaustive list of works surprisingly missing from this submission where a comparison and or delineation from would be very important:
  - Recently the line of research around SPEX [1] (followed-up by ProxySPEX [2] ) also identifies *sparse interactions* (the paper often refers to sparse interactions) based on the Möbius representation of the value function. Those sparse Möbius interactions are then transformed into the well-known Shapley Interactions. The paper often refers to those sparse interactions and how to find them. SPEX does so too.
   - The line of research on model-agnostic estimation of Shapley Interactions summarized in the `shapiq` [3] library. In the last years a couple of methods to estimate Shapley Interactions have been proposed that sample coalitions of players, evaluate the value function on these coalitions and based on this construct estimates of different Shapley interaction indices. Missing estimators that like this paper a based on the representation described in Fujimoto et al. (2006) are SHAP-IQ [7] and SVARM-IQ [8]. Notably, the work [4] presenting the faithful interaction index presents two estimation procedures as well, one related to KernelSHAP for the faithful interactions and one permutation estimator for the Shapley interaction index (similar to your permutation-based estimator). SPEX [1] and ProxySPEX [2] are also model-agnostic estimators. Most of the estimation methods presented here are also available for direct use in the shapiq library and it seems like they even integrated the new ProxySPEX method there recently.
   - Lastly, even if it is not the focus the different interaction indices should also be referenced and delineated from this work accordingly. The Shapley interaction index is already present in the paper but not properly compared to. Its efficient generalization [6] and the Shapley Taylor index [5] is missing. Lastly the faithful interaction index [4] is also not present, which is probably the most interesting angle for this work.
---
- **Weak Empirical Evidence:** As it stands currently the empirical side of this work is rather limited and leaves much to be desired. The main experiments are rather small case studies with a very limited sample size. Given that the work did not adequately compare itself to the most important related work it is also clear that the empirical evaluation of the method is rather limited in its current form. Of course it would be important to compare the interactions identified by this work with interactions derived from the different interaction indices and their estimations. A good selection of methods would be desirable given that most of the methods seem to be readily available in the shapiq library.
- **No Ablations:** The paper does not discuss the methods hyperparmaters nor does it show a proper analysis of its performance. I am unsure how efficient the method is in computing the interactions and how noisy its estimates are. A proper evaluation is needed. Questions that should be addressed: How does the method behave when the player size increase? What are the failure modes of the methods? What happens with no higher-order interactions and little higher order interactions (the literature [3,4,7,8] uses Sum of Unanymity Games for something like this)? What is the estimation quality of the estimator (when does it converge or not)?
- **No Code:** It's sad to see another paper not submitting the source code of their work for peer review. This greatly limits the reviewing quality and ultimately my view on this work.
---
- **Minor**: I had issues understanding Figure 1 and the central theorems for some time. I do not think Figure 1 in its current form helps all too much in explaining the core concepts since I am missing a lot of information in the infographic. I do not know how to easily improve this but maybe start of with giving the sets (R and T a good name and include some arrows what goes where or color code the formulas...)
- **Typos:** While reading I spotted some typos in the manuscript and small grammatical errors, but nothing major.

**Questions:**

- Q1: How does your interaction computation scheme scale? Is it quick? Is it slow? What are its direct failure modes?

---

> ### Author Response · Authors · 2025-11-26
>
> Thank you for the thoughtful and constructive feedback. We have carefully reviewed the suggested references and integrated their perspectives into our discussion. To clarify our motivation, contributions, and relation to prior work, we have strengthened Section 3.4 with an explicit proof of the connection to Harsanyi dividends, added a new Section 3.5 (Relation to Prior Work), and included several additional experiments in the Appendix. We would be grateful if you could also consider the overall narrative presented in our general response.
>
> **Q1. Relation to Prior Work**
>
> Thank you for pointing out the missing connections to recent work. We have carefully reviewed the suggested literature and have added a dedicated related-work section (Section 3.5 in the revised manuscript). Below we briefly summarize how our work relates to and differs from these prior studies.
>
> **Sparse Interactions.** Recent work has investigated scalable methods for measuring sparse high-order interaction attributions in large language models [1,2]. SPEX estimates a sparse Fourier representation of the value function and maps its coefficients to interactions [1]. ProxySPEX improves inference efficiency by training a gradient-boosted tree proxy and leveraging the hierarchical structure of LLM interactions [2]. These approaches provide strong empirical evidence that deep models often exhibit sparse, hierarchical interaction structures, and they highlight the importance of explicitly analyzing such interactions when interpreting model behavior. Our work is closely aligned with this perspective, but we focus on exploring those sparse coalitions to identify how the interaction effects affect Shapley-based attributions.
>
> **Shapley Interaction Index.** Our analysis starts from analyzing how the intrinsic interaction structure of non-additive and often non-convex models distorts Shapley-based explanation. To explicitly understand it, we decompose the marginal contribution into the consecutive sum of interaction effects, and then obtain the expectation of interaction terms in Theorem 2. Although derived from a distinct perspective, this term aligns with the standard characterization of interaction indices as weighted sums of discrete derivatives, as established in the classical Cardinal Interaction Index (CII) framework [3].
>
> The Shapley-Taylor Interaction Index (STI) [4] and Faith-Shap [5] both aim to produce interaction indices that satisfy efficiency. Since efficiency alone does not uniquely determine an interaction index, STI introduces the interaction-distribution axiom, whereas Faith-Shap views Shapley values as the best linear approximation and generalizes this principle to higher-order approximations for each subset. Bordt et al. [6] further introduce n-Shapley values and show that n-Shapley, STI, and Faith-Shap all recover GAM representations. All these methods recover Harsanyi-dividends when full cardinality of subsets is considered, but they differ in how they allocate or summarize high-order effects into the max-order terms. In our approach, since we gradually decompose marginal contributions from low to high order (as in Lemma 1), our max-order interaction term naturally accumulates all higher-order dividends. Thus, the second term in Theorem 2 corresponds to the aggregation of Shapley-Taylor interactions uniformly allocated to each feature in $i\cup R$.
>
> **Estimation.** The existing permutation-sampling baselines estimate all $k$-th order discrete derivatives for *every* subset, which leads to high computational cost. While each derivative evaluation in the per-sample complexity of our method is similar to STI [4], the overall strategy (utilized in Section 6) is different. Our approach expands interactions **incrementally** by measuring the variance of each coalition and selecting only those whose variance indicates meaningful higher-order effects. This avoids computing derivatives for the majority of $k$-order subsets, yielding substantial savings in sparse-interaction settings common in deep models. Also, STI first select features using a separate importance method (SV or IG), which may miss variables with near-zero expectation but high-variance interactions. Our variance-guided expansion naturally captures such cases.
>
> Finally, we note that efficient estimators such as SHAP-IQ and SVARM-IQ [7,8] provide a single-evaluation pipeline that updates all interaction terms simultaneously for CII-based interaction indices. Since our interaction term in Theorem 2 also follows CII structure, they may provide an efficient way to evalute the expectation/variance of our interaction terms. It would be great if we can utilize shapiq library [9] for further improving estimation efficiency.

---

> ### Author Response · Authors · 2025-11-26
>
> **Q2. Additional experiments, Scaling issue, and Failure Modes.**
>
> The permutation-based estimator in Theorem 3 is considerably more stable and accurate than the set-based estimator. We have conducted the additional experiments in Appendix F (Interaction Estimation Accuracy). It achieves lower estimation error with the same number of evaluations and converges more smoothly across multiple interaction orders.
>
> However, computing discrete derivatives itself for high-order interactions still remains computationally demanding. Since our formulation belongs to the Cardinal Interaction Index (CII) family, existing acceleration frameworks, such as SHAP-IQ, can potentially be applied to our approach. Integrating these techniques is a natural direction for future work and would allow our approach to scale more effectively to large feature sets.
>
> Most importantly, the strength of our pipeline lies in **reducing the number of candidate coalitions** before higher-order evaluation. Both Figure 4 (c) in the main paper and the additional experiment in Appendix D (Figure 6 in the revised manuscript) show that **only supersets derived from high-variance coalitions produce meaningful higher-order interactions**. Low-variance coalitions consistently yield negligible interaction magnitude, indicating that they require no further expansion. Thus, the variance criterion eliminates the vast majority of supersets from consideration. This pruning effect becomes especially powerful in models with sparse interaction structure, such as large language models [1]. In such settings, meaningful interactions are concentrated in a small number of feature groups, and our variance-guided expansion reduces the combinatorial search space dramatically.
>
> In addition, Figure 7 compares these variance-based findings with several existing interaction indices (SII, BII, STI, Faith-Shap) [3,4,5], all of which compute weighted expectations of discrete derivatives. These indices assign similarly small values to both families and therefore fail to distinguish their fundamentally different interaction behaviors, largely due to sign cancellation in expectation-based evaluations. In contrast, variance uniquely captures the context-sensitive higher-order effects that these indices overlook.
>
> One **potential failure mode** arises because variance quantifies the *instability* of interaction effects across contexts, but it does not distinguish whether those effects are beneficial or detrimental to the model’s prediction. In principle, a coalition $R$ may exhibit high variance because it consistently reduces the target logit in certain contexts. In such cases, variance-based pruning alone would not prevent us from expanding supersets that are ultimately unhelpful for the model’s decision.
>
> A practical diagnostic for this issue is to monitor the expected marginal contribution $\mathbb{E}_T[v(R\cup T)-v(T)]$ at each expansion step. This quantity indicates whether the coalition $R$ contributes positively or negatively to the model output, allowing us to avoid continuing along directions that do not support the prediction. Empirically, however, we observe that such “negative high-variance coalitions’’ are rare in typical image classifiers. We think that classifiers generally operate by *increasing* the logit of the target class rather than by actively suppressing others, so feature subsets with high variance tend also to increase the target logit.
>
> [1] Kang, Justin Singh, et al. "Spex: Scaling feature interaction explanations for llms." .In Forty-second International Conference on Machine Learning, 2025.
>
> [2] Butler, Landon, et al. "ProxySPEX: Inference-Efficient Interpretability via Sparse Feature Interactions in LLMs." *arXiv preprint arXiv:2505.17495.* 2025.
>
> [3] Grabisch, Michel, and Marc Roubens. "An axiomatic approach to the concept of interaction among players in cooperative games." *International Journal of game theory* 28.4. 1999.
>
> [4] Sundararajan, Mukund, Kedar Dhamdhere, and Ashish Agarwal. "The shapley taylor interaction index." *International conference on machine learning*. PMLR, 2020.
>
> [5] Tsai, Che-Ping, Chih-Kuan Yeh, and Pradeep Ravikumar. "Faith-shap: The faithful shapley interaction index." *Journal of Machine Learning Research* 24.94. 2023.
>
> [6] Bordt, Sebastian, and Ulrike von Luxburg. "From Shapley values to generalized additive models and back." *International Conference on Artificial Intelligence and Statistics*. PMLR, 2023.
>
> [7] Fumagalli, Fabian, et al. "SHAP-IQ: Unified approximation of any-order shapley interactions." *Advances in Neural Information Processing Systems* 36. 2023.
>
> [8] Kolpaczki, Patrick, et al. "SVARM-IQ: Efficient Approximation of Any-order Shapley Interactions through Stratification." *AISTATS*. 2024.
>
> [9] Muschalik, Maximilian, et al. "shapiq: Shapley interactions for machine learning." *Advances in Neural Information Processing Systems* 37. 2024.

---

### Official Review · Reviewer_qZTb · 2025-10-29

**Soundness:** 3
**Presentation:** 3
**Contribution:** 2
**Rating:** 4
**Confidence:** 3

**Summary:**

This paper proves that the Shapley value can be expressed as a weighted sum of its Shapley interaction terms, formalized through the Harsanyi dividends in game theory. The authors introduce a permutation-sampling strategy to estimate these interaction terms and provide a proof of its correctness. They further argue that features with high variance tend to contribute more through higher-order interactions, and propose using variance as a heuristic to select informative feature subsets for computing these interactions. In experiments on both vision and language models, they demonstrate that this filtering heuristic is effective: high-variance features tend to correspond to semantically meaningful regions, such as the central objects in images.

**Strengths:**

1. The paper is clearly written and well-structured, with experiments spanning a broad range of both vision and language models.

2. It introduces an alternative perspective on Shapley values by expressing them as a sum over interaction terms.

3. The authors propose a thoughtful feature filtering strategy based on variance, avoiding arbitrary feature selection when computing interaction terms.

4. This enables a practical “model auditing” approach, allowing users to explore and interpret predictions by focusing on high-variance feature subsets.

**Weaknesses:**

1. While I am familiar with SHAP and have some background in higher-order SHAP interaction terms, I am not an expert on this specific area of high-order interaction terms. One of the main limitations of the paper is the lack of a thorough discussion of related work on interaction-based SHAP explanations, which makes it difficult to assess the novelty of the proposed approach. In particular, it is unclear how the interaction terms that are analyzed here are related or differ mathematically from prior formulations such as (1) Shapley-Taylor interactions, (2) the SHAP-IQ framework, and (3) Shapley residuals. Given that there is a substantial body of work on SHAP interaction terms, the paper should better clarify how its proposed analysis relates or differs to these many existing approaches.


2. In addition to a thorough mathematical and formal comparison with existing interaction definitions, it would also be helpful to include an empirical comparison of these interaction terms to other interaction definitions to better assess the novelty and practical significance of the proposed approach.

3. The idea that Shapley values can be decomposed into a sum of higher-order interaction terms is not novel and has been previously established, for example, in Shapley-Taylor interactions derived from a Taylor series expansion.

4. It is unclear how much of the theoretical contribution is truly novel relative to existing results on Harsanyi dividends in the game theory literature, and the proofs provided appear relatively straightforward.

5. While using variance to filter important feature subsets is a reasonable strategy, the motivation behind what could be the use of computing these interactions remains somewhat unclear. A reader may still wonder: “so what?” Why is it meaningful to compute high-order interactions among high-variance features, and what practical insight does this provide? Although the results show that these subsets tend to align with the main objects in images, it is not evident what actionable or interpretive value this offers. Have you uncovered any interesting model behaviors or insights through this analysis that justify its usefulness?

6. It is unclear how the proposed estimation approach would scale to higher-order interactions beyond those evaluated in the experiments.

**Questions:**

1. How does your mathematical analysis differ from prior formulations of higher-order SHAP interactions, such as Shapley-Taylor, SHAP-IQ, and Shapley residuals, as well as other related notions?

2. What aspects of the connection between Harsanyi dividends and Shapley values are genuinely new? Was the decomposition of Shapley values via Harsanyi dividends already been established in the game theory literature? If this connection is already known, how does your work provide new theoretical insight?

3. In what way does the proposed mathematical formulation justify prioritizing features with high variance as a first step before computing higher-order interactions?

4. What is the practical benefit of the proposed pipeline: filtering high-variance features and then computing higher-order SHAP interactions? What downstream insights or applications does this enable?

5. How does the proposed estimation method scale to higher-order interactions? Are there guarantees or empirical evidence regarding computational efficiency or feasibility for large feature sets?

---

> ### Author Response · Authors · 2025-11-26
>
> Thank you for the thoughtful and constructive feedback. We have carefully reviewed the suggested references and integrated their perspectives into our discussion. To clarify our motivation, contributions, and relation to prior work, we have strengthened Section 3.4 with an explicit proof of the connection to Harsanyi dividends, added a new Section 3.5 (Relation to Prior Work), and included several additional experiments in the Appendix. We would be grateful if you could also consider the overall narrative presented in our general response.
>
> **Q1. How does your mathematical analysis differ from prior formulations of higher-order SHAP interactions, such as Shapley-Taylor, SHAP-IQ, and Shapley residuals, as well as other related notions?**
>
> Our representation in Theorem 2 stems from the observation that the marginal contribution used in Shapley computation, $\Delta_i v(S)$, can be **recursively decomposed** into a consecutive sum of interaction effects, as formalized in Lemma 1. We apply this decomposition because, in non-additive and especially non-convex functions common in deep networks, $\Delta_i v(S)$ can vary drastically with the context set $S$. For example, components such as max pooling or attention mechanisms, introduced in Section 4, can lead to unexpected (even negative) interaction effects between $i$ and the features in $S$, which may hinder model interpretation.
>
> To make these effects explicit, we extend each marginal contribution into an average k-th order interaction term in Theorem 2, which provides a summarization of higher-order interaction effects without requiring full Harsanyi decomposition. Structurally, this term resembles the formulation of the Shapley-Taylor index (STI)[1]. Actually, the interaction distribution axiom of STI gives all the higher-order interaction effects to the max-order term. Consequently, the second term in Theorem 2 corresponds to the aggregation of STI uniformly allocated to each feature appearing in $i\cup R$. However, in our analysis, we evaluate the interaction of $S$ at $s$-th order representations, rather than fixing max-order $k$ like STI. This avoids the limitation highlighted by Kumar et al. [2], where STI says nothing about whether the variables within $S$ interact once a player outside of $S$ is involved (when $s<k$). Furthermore, our framework in Section 6 gradually explores interaction effects from a single-feature evaluation to higher-order coalitions by extending feature subsets that potentially have an interaction with a context set (outside of S).
>
> Our work also connects to prior studies pointing out situations where Shapley values may fail to reflect proper feature importance, particularly when conflicting interaction signs cause attributions to collapse toward zero [2,3,4]. Kumar et al. [2] address this issue through Shapley Residuals, which quantify deviations from inessentiality. Such residuals can also be non-zero values when $R$ interacts with players outside of $R$ as described in Section 5 of [2]. One difference is that our variance-based analysis directly signals the presence of higher-order interactions involving a given coalition $R$. For example, if $Var_T[\Delta_{ab} v(T)]$ is large, our framework indicates that $\{a,b\}$ should be extended. If $\Delta_{ac}v(\emptyset)$ has a meaningful value, it should be detected through earlier variance evaluations on $\Delta_{a}v(T)$, not on $\Delta_{ab} v(T)$. In this way, our method naturally provides directionality on how to explore coalitions beyond the ones currently evaluated, rather than merely quantifying deviation from inessentiality.
>
> Many interaction indices belong to the Cardinal Interaction Index (CII) class, and SHAP-IQ [5] provides a highly efficient single-evaluation pipeline that updates all interaction terms simultaneously for this class. Since our interaction term in Theorem 2 also follows CII structure, SHAP-IQ may provide an efficient approach to sample and evaluate the expectation/variance of our interaction terms. Again, we want to emphasize that our goal is to question whether expectation-based attributions are fundamentally proper for model interpretation, and (if not) to identify which subsets should be prioritized when exploring higher-order interactions rather than accelerating each evaluation. I believe that this leads to a complementary viewpoint with current interaction studies to explore meaningful coalitions and to effectively evaluate their influences on model decisions.

---

> ### Author Response · Authors · 2025-11-26
>
> **Q2. What aspects of the connection between Harsanyi dividends and Shapley values are genuinely new? Was the decomposition of Shapley values via Harsanyi dividends already been established in the game theory literature? If this connection is already known, how does your work provide new theoretical insight?**
>
> The decomposition of a set function into Harsanyi dividends is well-established in cooperative game theory, and the fact that Shapley values can be expressed as weighted sums of these dividends is also classical. Our contribution is extending it to derive a decomposition of the Shapley value at an *arbitrary interaction order* by expressing the marginal contribution $\Delta_i v(S)$ as a consecutive sum of interaction effects (Lemma 1) and then characterizing the resulting expected interaction term (Theorem 2).
>
> The expected interaction term in Theorem 2 provides a summarization of higher-order Harsanyi dividends. This leads to the following explicit form:
> $$E_T[ \Delta_{i\cup R}v(T)]=\sum_{(i \cup R)\subseteq L \subseteq N}\frac{1}{k}\binom{l}{l-k}^{-1}\alpha_L$$
> where $l=|L|$ and $\alpha_L$ is a dividend for $L$. It is a precise weighted summarization of all higher-order dividends associated with supersets of $i \cup R$ (We have provided the proof in Appendix of the revised manuscript). This generalized decomposition offers two new insights:
>  - **It reveals how Shapley’s marginal contribution implicitly summarizes high-order dividends**, providing an interpretable partial reconstruction of Harsanyi’s full expansion without requiring enumeration of all orders.
>  - **It highlights the inherent limitations of expectation-based interaction measures**, since the same averaging that summarizes high-order dividends can also cancel out important effects when dividends change sign or depend strongly on context. Prior studies have empirically noted such failures of Shapley values [2,4], but our formulation gives a structural explanation for why these failures occur in an arbitary interaction order: the expectation operator compresses informative variability into a single number.
>
> **Q3. In what way does the proposed mathematical formulation justify prioritizing features with high variance as a first step before computing higher-order interactions?**
>
> The expectation form in Theorem 2 provides a compact summarization of all higher-order Harsanyi dividends. This formulation allows us to determine *when* such a summarization is reliable and *when* it becomes insufficient. Specifically, the variance term $Var_T[\Delta_Rv(T)]$, quantifies how sensitively the interaction of a coalition $R$ depends on the surrounding context $T$.
>
> Since each $\Delta_R v(T)$ can be decomposed into higher-order Harsanyi dividends, a **high variance** implies that **there exist subsets** $S$  with **non-negligible Harsanyi dividends** $\alpha_{S\cup R}$, which contribute to $\Delta_R v(T)$ only when the additional features $S$ are included in the context $T$. That is, it signals the presence of **context-dependent higher-order interactions** that cannot be captured by the expectation alone and therefore warrant further exploration.
>
> In contrast, when the variance is **low**, the discrete derivative $\Delta_R v(T)$ is nearly invariant to the choice of $T$, and the expectation $\mathbb{E}_T[\Delta_R v(T)]$ becomes an adequate and faithful summary. In this case, $R$ barely has higher-order interaction effects with the other features. Also, in our experiments , we observe the low-variance coalitions frequently correspond to background image regions or uninformative tokens in language models (Section 6, Appendix E), while high-variance coalitions consistently align with feature groups that form strong higher-order interactions.

---

> ### Author Response · Authors · 2025-11-26
>
> **Q4. What is the practical benefit of the proposed pipeline: filtering high-variance features and then computing higher-order SHAP interactions? What downstream insights or applications does this enable?**
>
> The practical benefit of our pipeline is that it enables a **targeted and scalable exploration of higher-order interactions**. Rather than evaluating all possible coalitions, which grows exponentially, we prioritize only those involving high-variance features. High variance of $\Delta_R v(T)$ indicates that there exist context-dependent and non-negligible dividends in supersets of $R$, meaning the coalition participates in meaningful higher-order structure that cannot be captured by expectation alone.
>
> Conversely, **low variance acts as a reliability indicator.** When $\Delta_R v(T)$ changes little across contexts, the expected interaction $\mathbb{E}_T[\Delta_R v(T)]$ becomes a stable and trustworthy summary of the coalition’s effect. Such low-variance coalitions typically correspond to background regions or uninformative tokens in deep neural networks (Section 6, Appendix E). This allows us to prune a large portion of the search space early.
>
> This variance-guided strategy is particularly effective for deep networks, where interactions are known to be sparse and only a small subset of feature combinations meaningfully affect the model’s output. By focusing exploration on high-variance coalitions, our pipeline efficiently follows the “interaction pathways” while discarding uninformative branches.
>
> **Q5. How does the proposed estimation method scale to higher-order interactions? Are there guarantees or empirical evidence regarding computational efficiency or feasibility for large feature sets?**
>
> First, our permutation-based estimator (Theorem 3) is substantially more efficient and stable than the set-based estimator (Theorem 2). As shown in the new empirical analysis in Appendix F (Interaction Estimation Accuracy), the permutation-based method converges more consistently to the ground-truth interaction values and achieves significantly lower estimation error for the same number of evaluations. This provides empirical evidence that our estimator remains feasible even as the interaction order increases.
>
> Second, we acknowledge that computing discrete derivatives itself for high-order interactions remains computationally demanding. However, since our formulation belongs to the Cardinal Interaction Index (CII) family, existing acceleration frameworks, such as SHAP-IQ, can potentially be applied to our approach. Integrating these techniques is a natural direction for future work and would allow our approach to scale more effectively to large feature sets.
>
> Most importantly, the strength of our pipeline lies in **reducing the number of candidate coalitions** before higher-order evaluation. Both Figure 4 (c) in the main paper and the additional experiment in Appendix D (Figure 6 in the revised manuscript) show that **only supersets derived from high-variance coalitions produce meaningful higher-order interactions**. Low-variance coalitions consistently yield negligible interaction magnitude, indicating that they require no further expansion. Thus, the variance criterion eliminates the vast majority of supersets from consideration. This pruning effect becomes especially powerful in models with sparse interaction structure, such as large language models [6]. In such settings, meaningful interactions are concentrated in a small number of feature groups, and our variance-guided expansion reduces the combinatorial search space dramatically.
>
> [1] Sundararajan, Mukund, Kedar Dhamdhere, and Ashish Agarwal. "The shapley taylor interaction index." *International conference on machine learning*. PMLR, 2020.
>
> [2] Kumar, Indra, et al. "Shapley Residuals: Quantifying the limits of the Shapley value for explanations." *Advances in Neural Information Processing Systems* 34, 2021.
>
> [3] Bordt, Sebastian, and Ulrike von Luxburg. "From Shapley values to generalized additive models and back." *International Conference on Artificial Intelligence and Statistics*. PMLR, 2023.
>
> [4] Chang, Wonjoon, Myeongjin Lee, and Jaesik Choi. "Rethinking Shapley Value for Negative Interactions in Non-convex Games." *The Thirteenth International Conference on Learning Representations*. 2025.
>
> [5] Fumagalli, Fabian, et al. "SHAP-IQ: Unified approximation of any-order shapley interactions." *Advances in Neural Information Processing Systems* 36. 2023.
>
> [6] Kang, Justin Singh, et al. "Spex: Scaling feature interaction explanations for llms." .In Forty-second International Conference on Machine Learning, 2025.

---

### Official Review · Reviewer_TyLg · 2025-10-31

**Soundness:** 3
**Presentation:** 2
**Contribution:** 2
**Rating:** 4
**Confidence:** 3

**Summary:**

This paper provides a novel theoretical reinterpretation of the Shapley value as not only a measure of individual contributions but also a structured decomposition of higher-order interactions among players (features). The authors derive a generalized formulation showing that computing the Shapley value is equivalent to decomposing the characteristic function into interaction terms of arbitrary order and evenly distributing each among involved players. They further propose an unbiased permutation-based estimator for practical computation and demonstrate that the variance of low-order interactions can serve as an indicator of hidden higher-order structures. Experiments on deep neural networks (e.g., ViT, VGG, BERT) validate the theoretical claims and show how this framework enhances the interpretability of black-box models.

**Strengths:**

1. Novel Shapley Value Decomposition: The paper introduces a method to decompose Shapley values into higher-order interaction terms, offering a deeper understanding of feature interactions, especially in deep learning models.

2. Theoretical and Practical Insights: The paper bridges theory and practice by offering a solid foundation for higher-order interactions in Shapley value attribution, particularly for deep neural networks.

3. Empirical Validation: The method is empirically validated on real-world models (VGG, ViT), showing how analyzing higher-order interactions improves model interpretability in complex decision-making tasks.

**Weaknesses:**

1. Some theoretical aspects lack empirical analysis. For example, Theorem 3 proposes an estimation method for Shapley values, but does not provide a rigorous analysis of its estimation error.

2. There is a lack of comparison with previous methods (such as Harsanyi dividend). Why is the method proposed in this paper more effective at revealing higher-order interaction effects? In what situations does it offer advantages?

3. The paper mentions using variance analysis to identify higher-order interaction effects but does not explore in-depth how these results match with the specific interaction terms in the theoretical analysis.

4. The experimental section of the paper does not clearly reveal how higher-order interaction terms directly affect the inference process of neural networks. The variance alone does not clearly explain the specific role of higher-order interactions in model inference, and the paper lacks a quantitative analysis of these effects during the inference process.

**Questions:**

1. How does the method proposed in Theorem 3 for estimating Shapley values perform in terms of estimation error, and could the paper include a more rigorous empirical analysis to evaluate its accuracy?

2. How does the proposed method compare with previous methods like Harsanyi dividend in revealing higher-order interaction effects, and under what conditions does it offer a clear advantage?

3. Can the paper further explore how the variance analysis results match with the specific higher-order interaction terms identified in the theoretical analysis?

4. Could the paper provide a more detailed quantitative analysis of how higher-order interaction effects directly impact the neural network's inference process, beyond just variance analysis?

---

> ### Author Response · Authors · 2025-11-26
>
> Thank you for the thoughtful and constructive feedback. To clarify our motivation, contributions, and relation to prior work, we have strengthened Section 3.4 with an explicit proof of the connection to Harsanyi dividends, added a new Section 3.5 (Relation to Prior Work), and included several additional experiments in the Appendix. We would be grateful if you could also consider the overall narrative presented in our general response.
>
> **Q1. How does the method proposed in Theorem 3 for estimating Shapley values perform in terms of estimation error, and could the paper include a more rigorous empirical analysis to evaluate its accuracy?**
>
> We have added a new empirical study comparing the estimation error of the set-based estimator (Theorem 2) and the permutation-based estimator (Theorem 3). We use a VGG model trained on ImageNet and segment each image into 15 superpixels using SLIC. For evaluation, we compute the *ground-truth* interaction values for selected feature subsets and measure the absolute error between these true values and the estimator outputs. The detailed results are explained in Appendix F of the revised manuscript.
>
> Across multiple interaction orders, we report estimation error curves as a function of the number of sampled sets/permutations. The results show that while the set-based estimator is highly sensitive to sampled context sets, resulting in unstable and slow convergence, **the permutation-based estimator converges more smoothly and consistently**, with much lower variance across repeated runs. For the same number of evaluations, the permutation-based estimator achieves substantially lower estimation error, indicating better sample efficiency.
>
> **Q2 & Q3. How does the proposed method compare with previous methods like Harsanyi dividend in revealing higher order interaction effects, and under what conditions does it offer a clear advantage? + Can the paper further explore how the variance analysis results match with the specific higher-order interaction terms identified in the theoretical analysis?**
>
> The decomposition of a set function into Harsanyi dividends is well-established in cooperative game theory, and the fact that Shapley values can be expressed as weighted sums of these dividends is also classical. Our contribution is extending it to derive a decomposition of the Shapley value at an arbitrary interaction order. By expressing the marginal contribution $\Delta_i v(S)$ as a consecutive sum of interaction effects (Lemma 1), we obtain an expected interaction term whose structure captures higher-order higher-order dividends (Theorem 2):
>
> $$E_T[\Delta_{i\cup R}v(T)]=\sum_{(i \cup R)\subseteq L \subseteq N}\frac{1}{k}\binom{l}{l-k}^{-1}\alpha_L$$ where $l=|L|$ and $\alpha_L$ is a dividend for $L$. This provides a precise weighted summarization of all higher-order dividends over supersets of $i\cup R$, and we include a complete proof in the revised Appendix.
> However, computing all Harsanyi dividends requires evaluating every coalition, making it exponential in dimension and often infeasible in deep learning settings. Our strategy offers the following clear advantages:
>
> (1) **Guidance for exploring higher-order coalitions via variance**
>
> The expectation form in Theorem 2 acts as a compact summarization for the full dividend expansion. It allows us to assess whether further exploration of higher-order coalitions is necessary. With a low variance of $\Delta_{i\cup R}v(T)$, higher-order dividends are either negligible or consistently aligned, so the expectation accurately summarizes their effect. A high variance of $\Delta_{i\cup R}v(T)$ indicates that dividends change sign or are highly context-dependent. In such cases, the expectation collapses important structure, which is also noted in prior work [2,3,4]. This variance of our interaction term becomes a signal that we must expand $i\cup R$ to identify the responsible higher-order interactions.
>
> In our empirical variance analysis (Section 5.2), we observe exactly this behavior. High variance consistently corresponds to higher-order dividends with substantial magnitude, and the associated features indeed participate in meaningful interactions with others.
>
> (2) **Efficiency in sparse interaction regimes**
>
> Sparse interaction structure is widely observed in deep networks [1]. In these settings, only a small fraction of higher-order coalitions have meaningful dividends, while the majority contribute negligibly. Computing dividends for all coalitions requires heavy cost on these negligible terms. In contrast, our variance-guided approach naturally focuses computation on the small number of coalitions that matter.

---

> ### Author Response · Authors · 2025-11-26
>
> **Q4. Could the paper provide a more detailed quantitative analysis of how higher-order interaction effects directly impact the neural network's inference process, beyond just variance analysis?**
>
>
> We address this question through the quantitative examples presented in Figures 5 (vision classifier, Section 5) and 6 (language classifier, Appendix). In these experiments, our variance-guided approach identifies the higher-order coalitions that are most likely to contribute meaningfully to the model’s prediction. For each such coalition, we report its **expected marginal contribution $\mathbb{E}_T[v(R\cup T)-v(T)]$**, which shows the actual increase in the model’s logit value when the coalition is added. It directly reflects how strongly the coalition impacts the network’s inference process. These results demonstrate that the coalitions discovered through our variance-based filtering are also functionally influential.
>
> To further strengthen this connection, we conducted an additional quantitative experiment (Figure 6 in Appendix of the revised manuscript), designed to evaluate whether high-variance subsets indeed form more meaningful higher-order coalitions than low-variance subsets. We construct two families of supersets:
>
> 1. **High-V** supersets that include high-variance features but exclude low-variance features;
> 2. **Low-V** supersets that include low-variance features but exclude high-variance features.
>
> The same grouping procedure is applied to second-order feature coalitions to construct candidate third-order coalitions.
>
> For each superset $R$, we measure the interaction magnitude $\mathbb{E}_T[|\Delta_R v(T)|]$, the interaction variance $\mathrm{Var}_T[\Delta_R v(T)]$, and its expected marginal contribution $\mathbb{E}_T[v(R\cup T) - v(T)]$. Our findings show that High-V supersets consistently exhibit much larger interaction magnitude and significantly higher variance, indicating substantial high-order structure. **These High-V supersets also have large marginal contributions**, demonstrating that they directly affect the model’s inference. In contrast, Low-V supersets show negligible interaction magnitude and negligible marginal contributions, confirming that they do not warrant further expansion.
>
> In addition, Figure 7 compares these variance-based findings with several existing interaction indices (SII, BII, STI, Faith-Shap) [5,6,7], all of which compute weighted expectations of discrete derivatives. These indices assign similarly small values to both High-V and Low-V families and therefore fail to distinguish their fundamentally different interaction behaviors, largely due to sign cancellation in expectation-based evaluations. In contrast, variance uniquely captures the context-sensitive higher-order effects that these indices overlook.
>
> Together, these results provide empirical evidence that variance serves as a reliable indicator for locating higher-order coalitions that genuinely affect the model output. Moreover, they validate that our variance-guided expansion navigates the combinatorial search space efficiently by pruning low-variance coalitions and focusing exploration on the high-variance ones.
>
>
> [1] Kang, Justin Singh, et al. "Spex: Scaling feature interaction explanations for llms." .In Forty-second International Conference on Machine Learning, 2025.
>
> [2] Kumar, Indra, et al. "Shapley Residuals: Quantifying the limits of the Shapley value for explanations." *Advances in Neural Information Processing Systems* 34, 2021.
>
> [3] Bordt, Sebastian, and Ulrike von Luxburg. "From Shapley values to generalized additive models and back." *International Conference on Artificial Intelligence and Statistics*. PMLR, 2023.
>
> [4] Chang, Wonjoon, Myeongjin Lee, and Jaesik Choi. "Rethinking Shapley Value for Negative Interactions in Non-convex Games." *The Thirteenth International Conference on Learning Representations*. 2025.
>
> [5] Grabisch, Michel, and Marc Roubens. "An axiomatic approach to the concept of interaction among players in cooperative games." International Journal of game theory 28.4, 1999.
>
> [6] Sundararajan, Mukund, Kedar Dhamdhere, and Ashish Agarwal. "The shapley taylor interaction index." *International conference on machine learning*. PMLR, 2020.
>
> [7] Tsai, Che-Ping, Chih-Kuan Yeh, and Pradeep Ravikumar. "Faith-shap: The faithful shapley interaction index." *Journal of Machine Learning Research* 24.94. 2023.

---

### Author Response · Authors · 2025-11-26
**General Response**

We sincerely thank the reviewers for their thoughtful and valuable feedback. Before addressing each comment individually, we would first like to clarify the motivation of our work, summarize our main conceptual contributions, and describe our approach more explicitly within the landscape of recent advances on Shapley-based interaction indices.

**Motivation.**

Although the Shapley value (SV) is defined through a weighted average of marginal contributions, the marginal contribution $\Delta_i v(S)$ can drastically change depending on the context set $S$, especially in non-convex functions typical of neural architectures. As illustrated in Section 4, operations such as *max* and *attention* can cause meaningful features to appear ineffective once certain representations are activated. These behaviors induce unexpected (sometimes negative) interaction effects between $i$ and $S$, and our work aims to explicitly analyze how such effects propagate into Shapley-based explanations.

**Decomposing Marginal Contributions.**

To make these hidden effects explicit, we observe that each marginal contribution $\Delta_i v(S)$ can be recursively decomposed into second-order interaction terms over permutations $\pi\in\Pi(S)$. These second-order terms can themselves be decomposed into third-order terms, and so on. Lemma 1 formalizes this into a general $k$-th order decomposition, revealing that **marginal contributions inherently summarize higher-order interactions through expectations over contexts**. Theorem 2 extends this result to the Shapley value itself.

**Implications for Complex Models.**

This decomposition explains behaviors observed in complex non-additive or non-convex models. As shown in Section 4 and Appendix C, discrete derivatives may change sign depending on the context, which is an issue also raised by Bordt et al. [1] and represented as the redundancy [2,3]. Our interpretation in Section 3.4 highlights why Shapley values may assign near-zero importance to features that are, in fact, highly influential: **expectation-based attributions average over heterogeneous interaction effects**, masking high-order behaviors and compressing them into a single number that can be misleading.

**Connection to Harsanyi Dividends.**

Our decomposition naturally bridges Shapley values (at $k=1$) and Harsanyi dividends (at $k=n$). While SV provides the compactest summarization without computing all dividends, which requires a heavy computational cost, this averaging can cancel out meaningful higher-order structure. By compromising both points, Theorem 2 suggests that the exact computation of dividends by $(k-1)$-th order and summarization for higher ones as the expected interaction on the context. This expected interaction term for the coalition $i\cup R$ summarizes all higher-order dividends for supersets of $i\cup R$ as follows:


$$E_T[\Delta_{i\cup R}v(T)] = \sum_{(i \cup R)\subseteq L \subseteq N} \frac{1}{k} \binom{l}{l-k}^{-1}\alpha_L$$

We include the full derivation in the revised Appendix.

---

> ### Author Response · Authors · 2025-11-26
>
> **Why Variance?**
>
> Variance plays a central diagnostic role in our framework.
>
> - When $\mathrm{Var}_T[\Delta_R v(T)]$ is **low**, $\Delta_R v(T)$ barely depends on the context $T$, and the expectation $\mathbb{E}_T[\Delta_R v(T)]$ accurately summarizes the interaction structure. Such coalitions empirically correspond to uninformative regions or tokens, which implies that they do not interact with other features and their near-zero attributions are reliable (Section 6 - Figure 4, Appendix - Figure 7).
> - When the variance is **high**, $\Delta_R v(T)$ depends strongly on context, indicating that the associated Harsanyi dividends fluctuate and that **substantial higher-order structure exists**. These are precisely the coalitions where expectation-based attributions fail and where further exploration is needed.
>
> Thus, variance provides a **direct and actionable signal** for refining the subset $R$, giving our method a principled path for discovering meaningful higher-order interactions.
>
> **Relation to Prior Work.**
>
> A substantial line of research has extended the classical Shapley value to quantify interactions with a specific axiom design, beginning with the Shapley Interaction Index (SII) [4]. Our analysis differs in that we start from the structural behavior of marginal contributions themselves, rather than selecting or designing an interaction index, and show how interaction effects distort Shapley-based explanations.
>
> Because our representation decomposes marginal contributions into consecutive orders, it inherently satisfies efficiency and aligns with several recent indices. The Shapley-Taylor Interaction Index (STI) [5] and Faith-Shap [6] both aim to produce interaction indices that satisfy the efficiency, in contrast to SII. Since the efficiency alone does not uniquely determine an interaction index, STI introduces the interaction-distribution axiom, whereas Faith-Shap views Shapley values as the best linear approximation and generalizes this principle to higher-order approximations for each subset. Both approaches extend SV to interaction indices and reduce to the original Shapley value at the single-feature level. Bordt et al. [1] further introduce n-Shapley values and show that n-Shapley, STI, and Faith-Shap all recover GAM representations. All these methods recover Harsanyi-dividends when full cardinality of subsets is considered, but they differ in how they allocate or summarize high-order effects into the max-order terms. In our approach, since we iteratively decompose marginal contributions from low to high order (as in Lemma 1), our max-order interaction term naturally accumulates all higher-order dividends. Thus, the second term in Theorem 2 corresponds to the aggregation of Shapley-Taylor interactions uniformly allocated across features in $i\cup R$. These interaction indices belong to the Cardinal Interaction Index (CII) class, and SHAP-IQ provides a highly efficient single-evaluation pipeline that updates all interaction terms simultaneously for this class [7].
>
> Our goal, however, is not to primarily propose another interaction index nor to accelerate their computation. Rather, we aim to highlight **when expectation-based attributions may be fundamentally unreliable** and to provide a **principled mechanism (via variance) for identifying the coalitions that merit higher-order exploration**. This complements and extends current interaction-index research by offering a diagnostic tool that targets meaningful coalitions directly.
>
> [1] Bordt, Sebastian, and Ulrike von Luxburg. "From Shapley values to generalized additive models and back." *International Conference on Artificial Intelligence and Statistics*. PMLR, 2023.
>
> [2] Fujimoto, Katsushige, Ivan Kojadinovic, and Jean-Luc Marichal. "Axiomatic characterizations of probabilistic and cardinal-probabilistic interaction indices." *Games and Economic Behavior* 55.1. 2006.
>
> [3] Chang, Wonjoon, Myeongjin Lee, and Jaesik Choi. "Rethinking Shapley Value for Negative Interactions in Non-convex Games." *The Thirteenth International Conference on Learning Representations*. 2025.
>
> [4] Grabisch, Michel, and Marc Roubens. "An axiomatic approach to the concept of interaction among players in cooperative games." *International Journal of game theory* 28.4. 1999.
> [5] Sundararajan, Mukund, Kedar Dhamdhere, and Ashish Agarwal. "The shapley taylor interaction index." *International conference on machine learning*. PMLR, 2020.
>
> [6] Tsai, Che-Ping, Chih-Kuan Yeh, and Pradeep Ravikumar. "Faith-shap: The faithful shapley interaction index." *Journal of Machine Learning Research* 24.94. 2023.
>
> [7] Fumagalli, Fabian, et al. "SHAP-IQ: Unified approximation of any-order shapley interactions." *Advances in Neural Information Processing Systems* 36. 2023.

---

### Meta-Review · Area_Chair_agWa · 2026-01-06

**Summary:**

This paper studies the higher-order interactions effects that can be hidden in the expected marginal contribution terms computed by Shapley-based approaches. To understand these better, the paper studies a higher order decomposition of these interactions and provides unbiased estimators for them. These are then used to uncover higher-order structures that are hidden from standard Shapley approaches.

This paper received four full reviews. The main concerns centered on (i) the positioning of this paper w.r.t to prior art that also studies high-order interactions in Shapley Values (including empirical comparisons); (ii) The precise connections between the higher-order terms and their variance-based approach; (iii) computational complexity aspects of their proposed methods.

I think these three main points were partially addressed: a related works section now positions in much better way the novelty of this contribution; some empirical results support the idea of using variance-based approaches to estimate higher-order interactions; and some general guidelines are given on potential accelerations. These improvements undoubtedly improved the paper, and I think this would put this paper at a borderline accept level. However, I believe that important questions raised by the reviewers remain and could improve the paper further, and I am not confident that any of the reviewers would have been very supportive of this paper if having been given the option to update their scores.

**Reviewer Concerns:**

### Rev TyLg
- This reviewer points to a lack of analysis of the estimation error of the proposed estimator. In the rebuttal, the authors provided an empirical analysis of the estimation error (comparing set-based and permutation based estimators). This addresses the concern - at least empirically.

- A lack of comparison to related work (Harsanyi dividends). The authors then provided a comment on connections to Harsanyi dividends, including a proof, and phrasing their Theorem 2 in these terms. There's an improvement due to avoiding the full computation of dividend, which is interesting.

- The proposed heuristics based on measuring variance might not relate tightly to their studied higher order interaction terms, and it's not clear how the higher order terms end up affecting the variance. Here, the authors added an experiment reporting the high-variance and low-variance coalitions, and comparing them to their interaction indices, arguing that high variance ones have higher marginal contribution. This point remains open, to my view, as there is still not a precise connection between their proposed heuristics and the higher order terms they are after.

### Rev qZTb
- Points to a lack of discussion of related works, mainly on Shapley-Taylor, ShapIQ, and Shap residuals, arguing a lack of novelty of analyzing higher order terms in Shapley values. The authors added a new Related Works section to better position their contribution that seems comprehensive. With this background, the novelty in the contribution is mainly on the evaluating interactions through studying variances.

- The practical benefits of computing these higher terms remains unclear or not convincingly motivated. The authors argue that the utility can be diagnostic, as shown in their examples in Fig 5 and 6. Still, this example is through their ad-hoc computed variance approach, and not new/unexpected failure modes/pathological cases were discovered based on their approach. Thus, this remains as a proof of concept.

- It remains unclear how this approach would scale to higher orders. In their rebuttal, the authors argue that their permutation test estimator is sample efficient and could be further accelerated (e.g. shapIQ).

### Rev z4D4
- Criticizes the lack of (many) related works (SPEX/ProxySPEX, shapiq, SHAP-IQ/SVARM-IQ, FaithSHAP, STI, efficient SII, etc).  This is now addressed by the Related Works section added (see above), and the focus on the variance-guided expansion, but this limits novelty. Furthermore, no empirical comparison was provided with many of the referenced methods.

- The empirical evaluation is limited to small case studies and lacks comparisons to interaction-index estimators and ablations. The authors provided new experiments that partly address these concerns (Appendix F).

- the lack of provided code is a negative factor. The authors have since added a discussion on scaling limits, explaining that higher-order terms remain expensive, and noting when their variance approach can fail.

- Lack of details on computational complexity and failure modes.

### Rev HZ32
- Also notes the lack of proper situations of the paper vis a vis prior work. The authors better explain how their method focuses on the decomposition of marginal contributions.

- Note the lack of novelty in their permutation based estimator (e.g. as in Faith-Shap). As a response, the authors note their variance-guided expanding as the main contribution.

- Missing a complexity analysis and computational comparison to other interaction methods. Here again, the authors mention that acceleration could be obtained with ShapIQ, but there's no rigorous computational analysis answer.

**Reviewer Scores:**

- Rev TyLg initially gave a 4, and might have increased to a 5 given the response.
- Similarly for qZTb
- Rev z4D4 provided a rejection recommendation, especially given the lack of proper framing w.r.t prior art. This has now been substantially addressed, and I thus anticipate that they would have improved their scores, but likely not to a level that would have recommended acceptance.
- HZ32 also provided a 4, and while they would have likely increased their score to 5, the partial reposes to the questions concerning computational complexity would have likely prevented the reviewer from increasing further.

---

### Decision · Program_Chairs · 2026-01-26

Reject